# Interactions of substrates and phosphinyl containing inhibitors with bacterial and human zinc proteases

Fatema Amatur Rahman[1], Imin Wushur[1], Ida Kristine Østnes Hansen[2], Tor Haug[2], Klara Stensvåg[2], Bibek Chaulagain[1], Tra-Mi Nguyen[1], Olayiwola Adedotun Adekoya[3], Nabin Malla[1], Jan-Olof Winberg[1], Ingebrigt Sylte[1,4]*

1 Pharmacology and Toxicology Research Group, Department of Medical Biology, Faculty of Health Sciences, UiT The Arctic University of Norway, Tromsø, Norway, 2 Norwegian College of Fishery Science, Faculty of Biosciences, Fisheries and Economics, UiT The Arctic University of Norway, Tromsø, Norway, 3 Centre for Pharmacy, Department of Clinical Science, Faculty of Medicine, University of Bergen, Norway, 4 Centre for Research and Education, University Hospital of North Norway (UNN), Tromsø, Norway

* ingebrigt.sylte@uit.no

## Abstract

Inhibitors of bacterial virulence is suggested to be a promising strategy in the fight against bacterial resistance. The zinc metalloproteases (MPs) thermolysin (TLN), pseudolysin (PLN, LasB) and aureolysin (ALN) are bacterial virulence factors from the M4 family of proteases structurally resembling human zinc MPs. Knowledge about the binding modes of substrates and inhibitors with the bacterial and human zinc MPs is therefore fundamental for developing inhibitors without strong off-target effects. In the present paper, we studied the molecular interactions and cleavage of TLN, the prototype enzyme of the M4 family, with the substrate Mca-Arg-Pro-Pro-Gly-Phe-Ser-Ala-Phe-Lys(Dnp)-OH (ES005) by matrix-assisted laser desorption ionization time-of-flight mass spectrometry (MALDI-TOF MS) and molecular dynamics (MD) simulations. Enzyme inhibition kinetic studies were used to test 5 compounds (H-1 - H-5) containing phosphinyl as the zinc binding group for their inhibition of TLN, PLN and ALN and of the human matrix metalloproteases MMP-9 and MMP-14. The MALDI-TOF MS results revealed that TLN cleaves ES005 at three sites. The most abundant cleavages were between Ala and Phe, and between Gly and Phe, while the third was between Ser and Ala. MD simulations with Ala or Gly in the $S_1$ subpocket and Phe in $S_1'$ subpocket gave stable interactions between ES005 and TLN. The MD simulations with Ser in the $S_1$ subpocket and Ala in the $S_1'$ subpocket showed larger movements of the substrate relative to the catalytic site than the others, which may explain that the Ser-Ala cleavage product was less abundant than the cleavage products from Ala-Phe and Gly-Phe. H-1 inhibited MMP-14 and MMP-9 with inhibition constants ranging from 0.89 to 30 µM but did not inhibit the bacterial zinc MPs. Induced fit docking showed that the aromatic group of H-1, that entered the $S_1'$

**Data availability statement:** All relevant data are within the manuscript and its Supporting Information files.

**Funding:** IS grant no: 1514-20, Funder: Health Authorities of Northern Norway (Helse Nord) URL: https://www.helse-nord.no/. The grant financed the experimental studies. FAR: PhD grant from UiT The Arctic University of Norway TH, KS, JOW. IS: Running costs from UiT The Arctic University of Norway.

**Competing interests:** The authors have declared that no competing interests exist.

subpocket of the human MMPs, is too big for the $S_1'$ subpocket of the bacterial zinc MPs. H-2 inhibited the human MMPs with inhibition constants ranging from 0.53 μM (MMP-9) to 3.0 μM (MMP-14) and the bacterial zinc MPs with inhibition constants ranging from 2.5 μM (TLN) to 80 μM (ALN). Induced fit docking indicated that H-2 interacted quite differently with the human and bacterial zinc proteases, but with primed and unprimed subpockets in both. H-3, H-4 and H-5 did not inhibit any of the zinc MPs with inhibition constants < 100 μM. The MD simulations of ES005 with TLN showed that the MALDI-TOF MS results could be explained by that a Phe in $S_1'$ subpocket generate more stable interaction with TLN than an Ala in that subpocket. The docking studies indicated that the size of the $S_1'$ subpocket is an important determinant for inhibitor selectivity between bacterial and human zinc MPs.

## Introduction

Metalloproteases (MPs) comprise a heterogenous family of proteases that use a metal ion to bind the substrate and polarize a water molecule to perform the hydrolytic cleavage. In the present paper we are focusing on the bacterial MPs thermolysin (TLN), aureolysin (ALN) and pseudolysin (LasB elastase, PLN) of the M4 family and the human matrix MPs (MMPs), MMP-9 and MMP-14, that belong to the M10 family of proteases [1]. All these enzymes utilize a zinc ion for hydrolytic cleavage. The zinc ion is coordinated with the polypeptide backbone through interactions with two histidines and a third amino acid, which is a histidine in the MMPs [2], and a glutamic acid in TLN, PLN and ALN [3]. Additionally, a general base is required, that accepts a proton from the polarized water molecule and transfers it to the nitrogen of the scissile bond. In both MMPs and bacterial M4 family enzymes, the general base is the glutamic acid in the conserved HExxH motif, while the histidines in this motif serve as the two zinc coordinating residues [3,4].

Human MMPs constitute a family of zinc MPs with important functions in many physiological processes and are overexpressed in various diseases. MMPs can have a dual role, either promoting disease initiation and progression, or preventing it [2,5–8]. They were once considered promising drug targets, and several compounds that to a certain extent selectively inhibited MMP activity were developed both by pharmaceutical companies and the academia [9–18]. However, clinical trials with MMP inhibitors were disappointing, resulting in a reduced research focus into MMPs as drug targets [19]. Based on recent progress in understanding their physiological role and their duality in different diseases, there has been a rebirth for the development of new MMP inhibitors as putative new therapeutics [19,20].

PLN, ALN and TLN are secreted bacterial virulence factors [3]. The use of inhibitors against virulence factors is a promising new strategy in the treatment of bacterial infections, either used alone or as an adjuvant to antibacterial drugs [21–23]. PLN is the principle extracellular virulence factor of the Gram-negative pathogen *Pseudomonas aeruginosa* [24]. This bacterium causes severe infections in the lungs, skin, eyes, soft tissues, urinary tract, and bloodstream, and is

the etiologic agent of chronic airway infections in patients with chronic bronchitis, bronchiectasis, and cystic fibrosis [24,25]. The pathogen was listed in the category "critical" on the World Health Organization's (WHO) priority list of bacterial pathogens for which research and development of new antibiotics are urgently needed [26]. ALN is secreted by the Gram-positive bacterium *Staphylococcus aureus*, one of the most harmful human pathogens, associated with methicillin resistance [27], that can cause serious infections if it enters the bloodstream or internal tissues. *S. aureus* is implicated in several different infections ranging from mild skin-related infections to more severe life-threatening and systemic infections such as bacteremia [28,29]. TLN is a thermostable protease of industrial use secreted by the Gram-positive bacterium *Bacillus thermoproteolyticus* [3]. The enzyme is the prototype of the M4 family of proteases, also termed the thermolysin family.

Most compounds that inhibit zinc MPs contain a zinc binding group (ZBG) in addition to a scaffold that interacts with subsites of the active site cleft. Well studied ZBGs include hydroxamic acid (HONH-CO), carboxylate ($CO_2^-$), thiolate ($S^-$) and phosphinyl ($PO_2^-$) [30,31]. Hydroxamic acid and thiolate are stronger ZBGs than carboxylate and phosphinyl [31,32]. The scaffold of inhibitors with hydroxamic acid, carboxylate, or thiolate as the ZBG targets the primed or unprimed subsites. However, X-ray crystallography studies show that compounds with the phosphinyl group may act as transition state analogues, enabling the connecting scaffold to interact with both primed and unprimed subsites of the active site. Such binding modes may potentially enhance the selectivity compared with inhibitors targeting primed or unprimed subsites, only [14,16,31–34]. Rouanet-Mehouas et al. [31] studied the contribution of different ZBGs to MMP binding by combining a structural scaffold with various ZBGs (phosphinyl, carboxyl, or hydroxamic acid). By using phosphinyl as the ZBG, they identified the MMP inhibitor RXP470, which is compound H-1 in the present study (Fig 1). They showed that different ZBGs affected both affinity and selectivity, as well as the inhibitor's position at the binding site [31]. RXP470 interacted with both primed and unprimed sites, whereas replacing the phosphinyl group with carboxylic acid or hydroxamic acid as the ZBG, along with the removal of a *p*-bromophenyl group, resulted in inhibitors that interacted only with the primed regions of the active site [31]. Binding strengths of inhibitors containing phosphinyl, carboxylic acids or thiolates as ZBG are pH dependent in contrast to those with a hydroxamic acid group [31,32].

In the present study we used matrix-assisted laser desorption ionization time-of-flight mass spectrometry (MALDI-TOF MS) and identified that TLN cleaves the substrate Mca-Arg-Pro-Pro-Gly-Phe-Ser-Ala-Phe-Lys(Dnp)-OH (ES005) at three different sites. The cleavages sites were starting points for molecular dynamics (MD) simulations of ES005 with TLN. These MD simulations were compared with simulations of the substrate Mca-Pro-Lys-Gly-Leu-Dpa-Ala-Arg-NH$_2$ (ES001) with MMP-9 and MMP-14. The aim of these simulations was to study the molecular interactions between the substrates and enzymes, and to identify amino acids in the primed and unprimed binding sites of the enzymes.

Inhibitors of bacterial virulence is suggested to be a promising strategy in the fight against bacterial resistance. However, inhibitors should not interfere strongly with human targets to be promising drug candidates. Human MMPs resemble the bacterial virulence factors of the M4 family in both structure and function. In order to identify structural determinants for selectivity between bacterial zinc MPs and human MMPs, we previously tested compounds with different ZBGs as inhibitors [3,35–39], including compounds with nitrogen as the donor atom for zinc chelation [40]. In this study, we also tested RXP470 (H-1) and four other compounds (Fig 1) with phosphinyl as the ZBG as inhibitors of MMP-9, MMP-14, TLN, PLN, and ALN, using the same substrates as in the MD simulations. Additionally, phosphinyl compounds were studied through induced fit docking into the zinc MPs.

## Materials and methods

### Materials

Dimethyl sulfoxide (DMSO), TRIS, calcium chloride (CaCl$_2$•2H$_2$O), sodium hydrogen phosphate (Na$_2$HPO$_4$), and sodium acetate (CH$_3$COONa) were from Merck (Darmstadt, Germany). Ethylenediaminetetraacetic acid (EDTA)

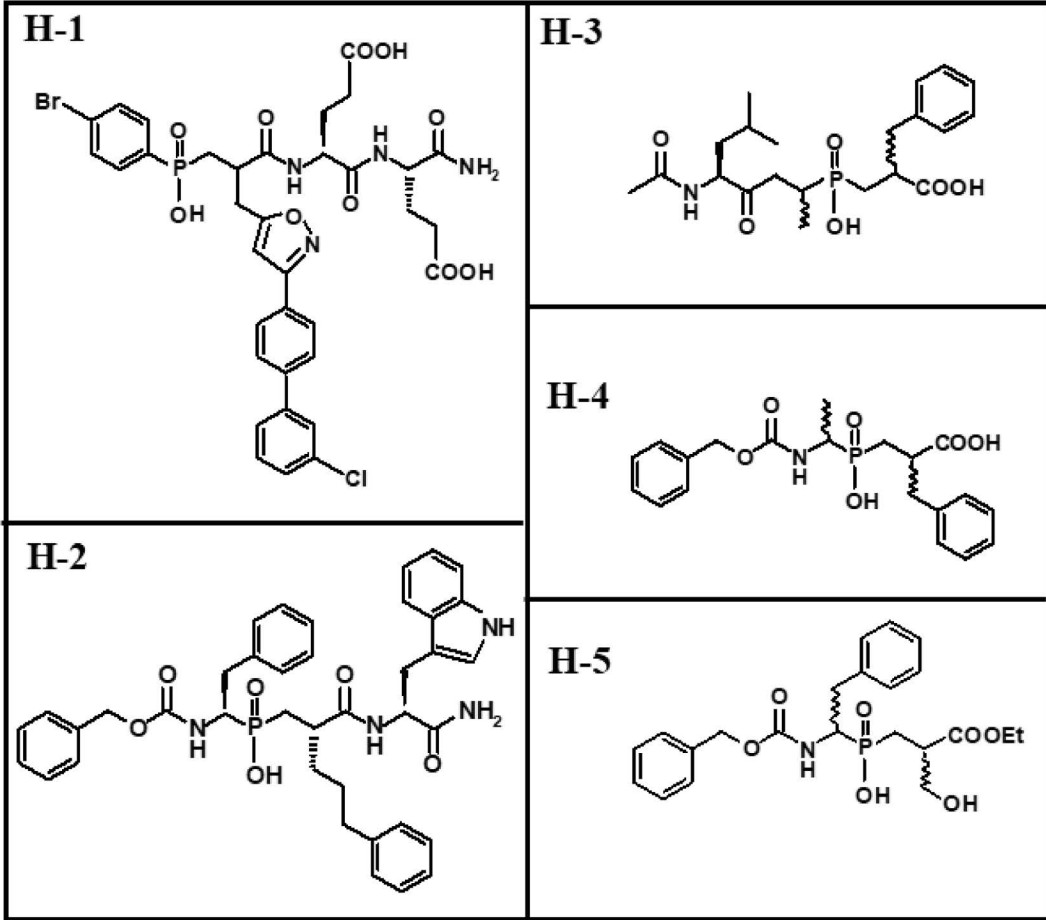

**Fig 1. H-compounds.** Molecular structures of the phosphinic compounds tested by enzyme inhibition kinetics.

was from Fluka (Buchs, Switzerland). Acrylamide, Coomassie Brilliant Blue G-250 and Triton X-100 were from BDH (Poole, UK). Hepes, Brij-35, silver nitrate, alkaline phosphatase-conjugated antibodies, p-aminophenylmercuric acetate (APMA), gelatin and α-Cyano-4-hydroxycinnamic acid (HCCA) were from Sigma-Aldrich (St. Louis, MO, USA). Magnetic trypsin beads (Mag-Trypsin) were purchased from Takera (Gothenburg, Sweden). Gelatin-Sepharose, was from GE Healthcare (Uppsala, Sweden). DC Protein Assay and unlabelled molecular weight standards were from BioRad (Richmond, CA, USA). Sf9 and High Five™ insect cells and Magic Marker molecular weight standards were from Invitrogen (Carlsbad, CA, USA). Western Blotting Luminol reagent and HRP-conjugated donkey anti-goat secondary antibody were from Santa Cruz (Santa Cruz, CA, USA). HRP-conjugated goat anti-rabbit secondary antibody was from Southern Biotech (Birmingham, AL, USA). Fetal bovine serum was from Biochrom AG (Berlin, Germany). Galardin (Gm6001), human MT1-MMP/MMP-14 (catalytic domain), TLN and PLN were from Calbiochem (San Diego, CA, USA), while ALN was from BioCentrum Ltd (Kraków, Poland). ES001 and ES005 were from R&D Systems (Minneapolis, MN, USA). Mca-PL-OH was from Bachem AG (Basel, Switzerland). Trifluoroacetic acid (TFA) and acetonitrile (MeCN) of MS quality were from VWR (West Chester, PA, USA). The Peptide Calibration Standard II was from Bruker Daltonik GmbH (Bremen, Germany). Synthesis of compounds H-1 - H-5 has been described in previous papers [32–34,41–44].

## Expression, purification and activation of recombinant human proMMP-9

The expression of recombinant full length proMMP-9 (rpMMP-9) in Sf9 and High Five insect cells and the purification were performed as previously described [38,45]. The amount of proMMP-9 was estimated spectrophotometrically at 280 nm using $\varepsilon_{280nm} = 114.36$ M$^{-1}$ cm$^{-1}$ [46] for the full length enzyme. Activation of the recombinant proMMP-9 was performed with APMA (auto-activation) and Mag-Trypsin as previously described [38], while the amount of active MMP-9 was determined by active site titration using galardin, also described previously [38].

## Enzyme inhibition studies of phosphinic compounds

**Determination of reaction velocities.** The substrate ES001 was used for APMA activated MMP-9 (MMP-9(A)), Mag-Trypsin activated MMP-9 (MMP-9(MT)) and MMP-14, while the substrate ES005 was used for ALN, PLN and TLN. The reaction velocities/initial rates ($v$) were determined at 37°C, at an excitation wavelength of 320 nm and an emission wavelength of 405 nm with a slit width of 10 nm using either a Perkin Elmer LS 50 Luminescence spectrometer and the FL WinLab Software Package (Perkin Elmer) or a Clario Star micro plate reader (CLARIOstar® BMG LABTECH).

The phosphinic compounds were dissolved in 100% DMSO to a final concentration of 10 mM. In all assays, a fixed substrate concentration of 4.0 μM was used in a total volume of 100 μL 0.1 M Hepes pH 7.5, 10 mM CaCl$_2$, 0.005% Brij-35 and 1.0% DMSO. Fixed enzyme concentrations were as follows; 0.05 nM MMP-9(A), 0.05 nM MMP-9(MT), 1.0 nM MMP-14, 1.4 nM ALN, 0.5 nM PLN and 0.21 nM TLN. Time dependent inhibitory experiments, with and without 100 μM of phosphinic compounds, were performed as follows: Compounds were either pre-incubated with proteases for 15 and 30 minutes at room temperature before initiating the enzymatic reaction by adding the substrate, or not pre-incubated, meaning the reaction was initiated by adding the protease to the mixture of substrate and compound. Control tests without the compound present were performed under the same conditions. Reactions were monitored for 30 minutes at 37°C.

**Determination of $K_m$ values.** $K_m$ value was determined for ES001 with Mag-Trypsin activated recombinant MMP-9 (MMP-9(MT)). Substrate concentrations used were 1–10 μM in a total volume of 100 μL of 0.1 M Hepes pH 7.5 containing 10 mM CaCl$_2$, 0.005% Brij-35 and 1.0% DMSO. Substrate concentrations above 10 μM resulted in quenching as reported previously [37]. Initial rate experiments were performed as described above to determine enzyme reaction velocities using a LS 50 Luminescence Spectrometer and the FL WinLab Software Package (Perkin Elmer). The $K_m$ and s.d values were calculated with Graph Pad Prism 5 using the Michaelis-Mention equation. The $K_m$ values of ES001 for MMP-9(A) and MMP-14, as well as those of ES005 for TLN, PLN and ALN were previously determined under the same experimental conditions as in the present work [35]. During these experiments, reactions were monitored for only one minute, and initial reaction velocities were determined from the activity curves.

**Determination of $IC_{50}$ and $Ki$ values.** Determination of inhibitory constants ($IC_{50}$) of the phosphinic compounds were performed with concentrations ranging from $10^{-10}$ to $10^{-4}$ M in the assay or eight concentrations resulting in 10–90% inhibition. A fixed substrate concentration and the same enzyme and buffer conditions as described above were used. Enzymes, with or without inhibitors, were either pre-incubated for 30 minutes at room temperature or not pre-incubated, and initial rate assays were performed as described above. Assays were performed using a Clario Star microplate reader. The $IC_{50}$ values with corresponding s.d. were calculated with GraphPad Prism 5 using equation 1 or 2, depending on the concentration range of the phosphinic compounds.

$$\frac{v_i}{v_0} = \frac{1}{\left(1 + 10^{(pIC50-pI)}\right)}$$

(1)

$$\frac{v_i}{v_0} = \frac{1}{\left(1 + \frac{[I]}{IC_{50}}\right)}$$

(2)

In equations 1 and 2, $v_i$ represents the enzyme activity in the presence of the inhibitor, while $v_0$ denotes activity in its absence, pI = -log [Inhibitor] in M and p$IC_{50}$ = - log $IC_{50}$ in M. All experiments were performed at least in triplicate. Equation 3 describe the relationship between $IC_{50}$ and $K_i$ values for substrate competitive inhibitors, considering the fixed substrate concentration and the enzyme's $K_m$ value for the substrate:

$$IC_{50} = K_i \left(1 + [S]/K_m\right)$$

(3)

**Quenching experiments.** Quenching experiments were conducted as previously described [37], to assess the extent to which the compounds quenched the time dependent enzymatic increase in fluorescence of the processed substrate products. Briefly, fluorescence measurements ($\lambda_{ex}$ = 320nm, $\lambda_{em}$ = 405 nm, slit width = 10 nm) at various concentrations of the Mca-fluorescent product (0–100 nM McaPL-OH) of ES001 and ES005, were determined in absence and presence of various concentrations of the phosphinic compounds (0–100 µM). Primary and secondary plots were used to determine if these compounds quenched the McaPL-OH fluorescence.

## MALDI-TOF mass spectrometry

MALDI-TOF MS spectra were recorded for samples containing different enzyme and substrate concentrations at various time points using an Autoflex Speed instrument (Bruker Daltonik GmbH, Leipzig, Germany) equipped with a 355-nm nitrogen laser for desorption and ionization. The ionization matrix, HCCA, was dissolved to 10 mg/mL in a standard solvent (50% MeCN, 47.5% MQ-water, and 2.5% TFA) by vigorous vortexing. Prior to the MALDI run, a main buffer (0.1 M Hepes, pH 7.5, 10 mM $CaCl_2$) was used to dilute TLN to stock concentrations of 1000, 100 and 50 nM, ES005–500 µM, and galardin to 10 µM. All stocks solutions were kept on ice. For the 0 min incubation, 1 µL TLN and 1 µL galardin were preincubated at 37°C in 8 µL of the main buffer for 10 min before adding 1 µL of ES005. For all other time points, 1 µL of ES005 was added to tubes containing 8 µL buffer and 1 µL of TLN at different concentrations, followed by incubation at 37°C. At each timepoint (0, 1, 3, 5, 10, 20, 30, 60, 120 min), 1 µL of galardin was added to stop enzyme activity. After the final incubation, 1 µL aliquots from each timepoint were mixed with an equal volume of HCCA and spotted onto a MALDI plate. Additional spots on the plate were assigned for the instrument Peptide Calibration Standard II (mass range from 700–3200 Da) and blank controls containing only buffer and HCCA. Samples were dried at room temperature before loading into the MALDI-TOF MS. Spectra were acquired at a laser power of 100 mV in reflector mode with a mass to charge (m/z) range of 500–1700. The resulting mass spectra were analysed using flexAnalysis software (v3.3, Bruker).

## Molecular modelling

Molecular modelling was performed using the Schrödinger 2021−2 software suit (Schrödinger, Inc., New York, USA, 2021). The Desmond program [47] ([https://www.deshawresearch.com/publications/Desmond-GPU_Performance_April_2021.pdf](https://www.deshawresearch.com/publications/Desmond-GPU_Performance_April_2021.pdf)) version 6.7 running on the graphics processing unit (GPU) of our workstations was used for MD simulations of ES001 with MMP-9 and MMP-14, and of ES005 with TLN.

**Induced fit docking of H-1 and H-2.** Induced fit docking of H-1 and H-2 (Fig 1) was performed within the active site cleft of MMP-9, MMP-14, TLN, ALN and PLN using the induced fit docking protocol. The 2D structures of H-1 and H-2 were drawn and converted to 3D using the 3D builder tools of Maestro 12.8. Different enantiomeric states were generated and optimized with the LigPrep module using the OPLS4 force field [48]. The obtained low energy conformations were used for induced fit docking. The following X-ray crystal structures were selected for docking: MMP-9 (PDB ID: **2OVZ**), a Glu402Gln mutant complexed with a phosphinate inhibitor; MMP-14 (PDB ID: **1BQQ**), in complex with TIMP-2 (an X-ray structure of MMP-14 with a small molecular inhibitor is not available in the PDB-database); PLN (PDB ID: **3DBK**), in complex with the inhibitor phosphoramidone; TLN (PDB ID: **1TLP**), in complex with a phosphoramidate inhibitor; and ALN (PDB ID: **1BQB**), representing the enzyme without inhibitor (an ALN-inhibitor complex was not available in the PDB-database).

Gln402 in the MMP-9 structure was mutated back to Glu402, as in the wild type MMP-9. Enzyme structures were protonated at pH 7.4 ± 0.2 using the Epik tool, missing hydrogen atoms were added, and structures were optimized with the OPLS4 force field [48]. An atomic distance of 12 Å from the centre of the co-crystallized inhibitors was used to generate the 3D grid box for docking giving a grid box volume of 24 Å x 24 Å x 24 Å. For PDB structures lacking a small molecular inhibitor (MMP-14 and ALN), the catalytic zinc atom served as the centre of the 3D grid box. Induced fit docking was performed for the H-1 and H-2 using default parameters and standard sampling, retaining 20 poses of each ligand. During induced fit docking, the ring conformations were sampled in an energy window of 2.5 kcal/mol. Non-planar conformations for amide bonds were penalized. The enzyme and the ligands Van der Waals scaling was set to 0.50. Residues within 5.0 Å of ligand poses were refined, and side chains were optimized. The induced fit docking scores were calculated to rank the different poses.

**Construction of enzyme-substrate complexes for MD.** The X-ray structure of an inactive MMP-9 mutant (Glu227Ala, corresponding to Glu402Ala in this manuscript) complexed with a substrate probe molecule (PDB ID: **4JIJ**), along with an X-ray structure of MMP-9 containing the prodomain, catalytic domain, and three FnII (fibronectin type II) modules (PDB ID: **1L6J**), were used as starting points to generate MMP-9/ES001, MMP-14/ES001, and TLN/ES005 complexes. The prodomain was deleted from the 1L6J structure, resulting in Phe107 as the new N-terminal residue, as observed in trypsin and MMP-3 activated MMP-9 [49].

The substrate ES001 has the same length as that of the substrate probe (Fig 2), with the only structural difference being the substitution of a *para* iodinated Phe in the probe [50] with a Leu in ES001. The iodinated Phe occupies the $S_1'$ binding cavity of MMP-9 [50], and in addition, MMP-7 cleaves the Gly-Leu bond in ES001 [51] indicating the Leu most likely resides the $P_1'$ position (Fig 2). Crystallographic water molecules were removed from both structures, and an initial MMP-9 - ES001 complex was constructed from the MMP-9-substrate probe complex (PDB ID: **4JIJ**) by mutating the iodinated Phe in $P_1'$ position to Leu. This MMP-9-ES001 complex was then superimposed onto a modified 1L6J structure (without the prodomain), allowing ES001 to be transferred into the catalytic site of the modified 1L6J structure. The 1L6J structure was chosen for MD simulations due to the influence of its three FnII domains on structural dynamics.

An initial MMP-14-ES001 complex was constructed by superimposing the X-ray structure of MMP-14 (PDB ID: **1BQQ**), with crystallographic water molecules removed, onto the generated MMP-9-ES001 complex (modified 1L6J). The ES001 structure was then transferred into the active site cleft of MMP-14 in a position similarly to the position in MMP-9.

Constructing TLN complexes with ES005 was more challenging due to the greater length of ES005 compared to ES001 and the substrate probe (Fig 2). This made direct generation of the ES005 structure from the substrate probe problematic. The X-ray structure of MMP-9 with the substrate probe (PDB ID: **4JIJ**) was superimposed with the X-ray structure of TLN (PDB ID: **1TLP**), without crystallographic water molecules, and the substrate probe was transferred into the TLN structure. ES005 was then drawn, converted to 3D using the 3D Maestro 12.8 3D builder tools, and optimized with the LigPrep module using the OPLS4 force field [48]. MALDI-TOF MS studies revealed that TLN cleaves ES005 at three sites: between Gly and Phe, between Ala and Phe, and between Ser and Ala (Fig 2). Three complexes of ES005 with TLN were therefore constructed, by manually positioning ES005 on top of the substrate probe within the TLN structure. For each complex, the $P_1'$ residue of the cleavage site (Phe, Phe and Ala, respectively) was positioned in the $S_1'$ subpocket of TLN. In the two complexes where Phe occupied the $S_1'$ subsite, it was superimposed onto the 4-iodo-phenylalanine of the substrate probe. In the third complex, where Ala occupied the $S_1'$ subpocket, its backbone atoms and Cβ-atom of Ala were superimposed with the corresponding atoms of 4-iodo-Phe of the substrate probe. Manual adjustments of ES005 side chains were necessary to prevent structural clashes in all three complexes.

The enzyme structure complexes of ES001 with MMP-9 and MMP-14, as well as the three TLN-ES005 complexes, were protonated at pH 7 ± 2 using the Epik tool. Missing hydrogen atoms were added, and complexes were optimized using the OPLS4 force field [48].

## MMP-9 substrate probe

## ES001

## ES005

**Fig 2. Substrate structures.** Two dimensional (2D) illustrations of the substrate probe from the X-ray crystallographic complex with MMP-9 (PDB ID: **4JIJ**), as well as the substrates Mca-Pro-Leu-Gly-Leu-Dpa-Ala-Arg-NH₂ (ES001) and Mca-Arg-Pro-Pro-Gly-Phe-Ser-Ala-Phe-Lys(Dnp)-OH (ES005). ES001 was used in enzyme inhibition studies with MMP-9 and MMP-14, while ES005 was used in studies with TLN, PLN, and ALN. The $P_1$ and $P_1'$ residues in MD simulations of ES001 with MMP-9 and MMP-14, and of ES005 with TLN, are indicated in the figure.

**Molecular dynamics simulations.** The five energy optimized substrate-enzyme complexes were used as starting points for 200 ns of MD simulations. The OPLS4 force field [48] was also used for MD simulations. Each molecular system was solvated in an orthorhombic box by using the TIP4P water model with a region of 10 Å between the surface of the enzyme-substrate complexes and the box walls. Chloride ions were added to neutralise the system. Equilibration of the molecular systems was performed using the default Desmond heating and relaxation protocol, which includes 100 ps of Brownian dynamics at constant temperature (10 K) and pressure, with restraints applied to solute heavy atoms. This was followed by a gradual heating process from 0 to 300 K. A cutoff radius of 9 Å was applied for short range electrostatic and van der Waals interactions, while long range electrostatic interactions were treated using the smooth particle mesh Ewald method [52]. Heating was followed by an NPT equilibration period such that the pressure and density were allowed to fluctuate as the system relaxed toward equilibrium. During these steps restraints were used on solute heavy atoms for 24 ps, while the last 24 ps were without atomic restraints. After convergence, MD simulations for 200 ns were performed under isothermal-isobaric ensemble (NPT) (temperature: 300 K, pressure: 1.01325 bar) with time steps of 2 fs. A Nose-Hoover thermostat and the Martyna-Tobias-Klein barostat were used to maintain constant temperature and pressure [53]. Coordinates were written every 100 ps, giving all together 2000 frames in each trajectory.

## Results and discussion

### Enzyme activity of the five MPs against ES001 and ES005

Different substrates are available for experimental studies with zinc MPs. Already in 1992, Knight and coauthors reported the synthesis of ES001 and its use as a sensitive substrate for MMP-1, MMP-2, MMP-3 and MMP-7 [51]. They showed that MMP-7 cleaved the Gly-Leu bond. Later it was also shown to be a very sensitive substrate for MMP-9 [54], and we therefore started to use ES001 as substrate for MMP-9. Later we studied the activation of MMP-2 by MMP-14 and successfully used ES001 as a substrate for MMP-14 [55]. In 2000, the bradykinin like substrate ES005 was developed as a sensitive substrate for the endothelin-converting enzyme-1 (ECE-1) [56]. The only differences between bradykinin and ES005 are amino acids 7 and 9, which is Pro and Arg in bradykinin, and Ala and Lys(Dnp) in ES005 (Fig 2). ECE-1 cleaved ES005 between Ala-Phe. TLN was also shown to cleave a modified bradykinin (Phe[5] (4-nitro)-bradykinin) between Pro[7]-Phe[8] [57]. TLN is known to prefer hydrophobic and/or bulky amino side chains in the $S_1$' subpocket [3] indicating that TLN at least has 3 putative cleavage sites in ES005 (Fig 2). In several of our previous studies, we have therefore used ES001 as a substrate for human MMPs, while ES005 has served as a substrate for bacterial virulence factors [35,38,40]. However, it has not been demonstrated that the bacterial enzymes prefer ES005 over ES001 and that the human MMPs favour ES001 over ES005.

To compare the activity of these enzymes against ES001 and ES005 assays were conducted using the same assay condition for all enzymes. The differences from the testing of phosphinic compounds described in the Material and Methods section were that DMSO was omitted, and the concentration of both substrates was 10 µM. A Perkin Elmer LS50 Luminescence Spectrometer was used to measure the activity. Reactions were monitor for one minute, and initial reaction velocities were determined from the activity curves. For each enzyme, the reaction with the presumed optimal substrate was performed first at an arbitrary enzyme concentration, followed immediately by the reaction with the alternate substrate using the same enzyme concentration.

The results demonstrated that the three bacterial enzymes had higher activity against ES005 than against ES001 (Table 1). TLN showed approximately 13-fold higher activity for ES005, while PLN and ALN displayed approximately 15-fold and 5-fold higher activity, respectively. However, the human enzymes showed a preference for ES001 over ES005. MMP-9(A) had around 15-fold, MMP-9(MT) approximately 20-fold, and MMP-14 nearly 29-fold higher activity against ES001 than ES005. Based on these findings, we used ES001 as the substrate for enzyme inhibition studies with MMP-9 and MMP-14, and ES005 for inhibition studies with TLN, PLN, and ALN as in our previous studies.

**Table 1. Enzymatic activity (ΔF/min). Enzymatic activity of bacterial (TLN, PLN and ALN) and human (MMP-9(A), MMP-9(MT), and MMP-14) enzymes against ES001 and ES005 (n = 4 in all assays).**

| Enzyme | Substrate | |
|---|---|---|
| | ES001 | ES005 |
| TLN | 27 ± 3 | 326 ± 36 |
| PLN | 9 ± 1 | 138 ± 4 |
| ALN | 65 ± 8 | 354 ± 61 |
| MMP-9(A) | 77 ± 5 | 5.1 ± 0.2 |
| MMP-9(MT) | 363 ± 22 | 18 ± 8 |
| MMP-14 | 230 ± 19 | 8 ± 2 |

## MALDI-TOF MS analysis of ES005 cleavage sites

TLN is known to prefer hydrophobic amino acids within the $S_1'$ subpocket [3], but the exact cleavage sites of ES005 (Fig 2) remain unidentified. To determine these sites, we performed a MALDI-TOF MS analysis. No TLN cleavage products were detected in the MALDI-TOF MS spectra immediately after incubating 50 µM ES005 (0 min incubation) with 10 nM TLN (Fig 3). The calculated m/z [M + H]+ value of ES005 is 1388.59 and the peak was observed at 1388.62, verifying the molecular weight of ES005 to be 1387.61 Da. At time zero, several peaks with slightly lower molecular masses were also observed, likely corresponding to the loss of O, $O_2$, NO and $NO_2$ due to fragmentation of the 2,4-dinitrophenyl group (Dnp) of the substrate (S1 Table). Furthermore, a peak observed of around m/z 1221.4 [M + H]+ is also likely a MALDI-induced fragment of ES005, caused by complete loss of Dnp from the substrate (Fig 3).

After 5 minutes of incubation with 10 nM TLN, three distinct cleavage products (m/z peaks representing [M + H]+ ions) of ES005 were observed (Fig 3, Table 2). Based on the observation of N-terminal sequences, ES005 was cleaved at three sites: Gly-Phe (m/z 642), Ser-Ala (m/z 876), and Ala-Phe (m/z 947). The most dominant cleavage site was Ala-Phe, followed closely by Gly-Phe, while Ser-Ala cleavage was less abundant (Figs 3–4). This cleavage pattern was observed at 1 minute with 5 nM TLN (Fig 4, S1 Fig in S1 File) and remained up to 30 minutes with 5 nM TLN and 5 minutes with 10 nM TLN (Figs 3–4, S1 Fig in S1 File). Longer incubation at both TLN concentrations gave increased cleavage at the Gly-Phe site relative to the Ala-Phe site. This effect was more pronounced when the full-length ES005 substrate was almost or completely digested (S1 Fig in S1 File, Fig 4). These findings suggests that the Ala-Phe cleavage product (Mca-Arg-Pro-Pro-Gly-Phe-Ser-Ala) was further cleaved to yield the Gly-Phe cleavage product (Mca-Arg-Pro-Pro-Gly). One µM galardin

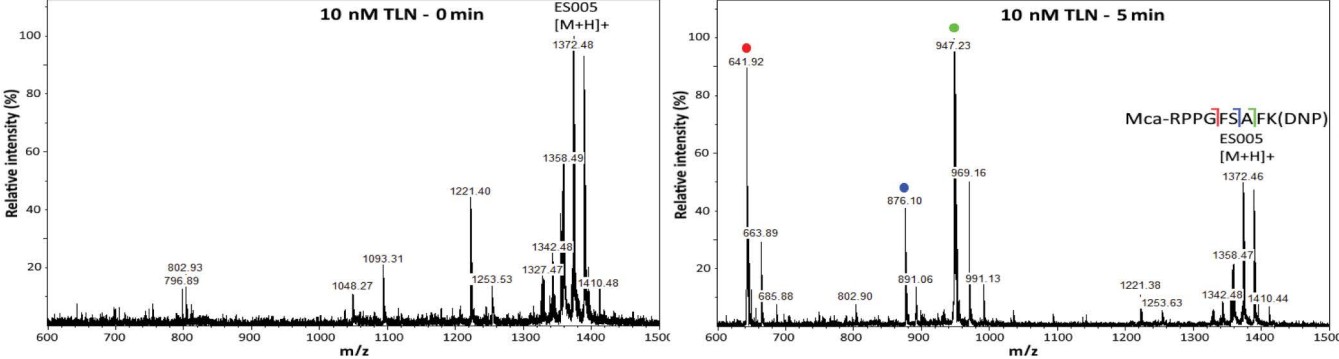

**Fig 3. MALDI-TOF MS spectra.** MS spectra obtained after 0 and 5 minutes incubation of 50 µM (Mca-Arg-Pro-Pro-Gly-Phe-Ser-Ala-Phe-Lys(Dnp)-OH) ES005 with 10 nM thermolysin (TLN). The m/z values marked in green, blue and red represents N-terminal cleavage products of ES005 obtained after Ala-Phe, Ser-Ala, and Gly-Phe cleavage, respectively.

**Table 2. MALDI-TOF MS data obtained after TLN cleavage of ES005.** The m/z values (representing [M + H]⁺ ions) of ES005 and the three observed cleavage products are shown. The observed m/z values represent the average value from all measurements performed.

| Substrate/fragment sequence | Formula | Calculated m/z [M + H]⁺ | Observed m/z [M + H]⁺ |
|---|---|---|---|
| Mca-Arg-Pro-Pro-Gly-Phe-Ser-Ala-Phe-Lys(Dnp)-OH (ES005) | C66H81N15O19 | 1388.59 | 1388.62[1] |
| Mca-Arg-Pro-Pro-Gly-Phe-Ser-Ala-OH (AF cleavage) | C45H58N10O13 | 947.43 | 947.42[2] |
| Mca-Arg-Pro-Pro-Gly-Phe-Ser-OH (SA cleavage) | C42H53N9O12 | 876.39 | 876.35 |
| Mca-Arg-Pro-Pro-Gly-OH (GF cleavage) | C30H39O9N7 | 642.29 | 642.20[3] |

[1] The [M + Na]⁺ ion (m/z 1410.58) was also observed.

[2] The [M + Na]⁺ ion (m/z 969.39) was also observed.

[3] The [M + Na]⁺ ion (m/z 664.18) was also observed.

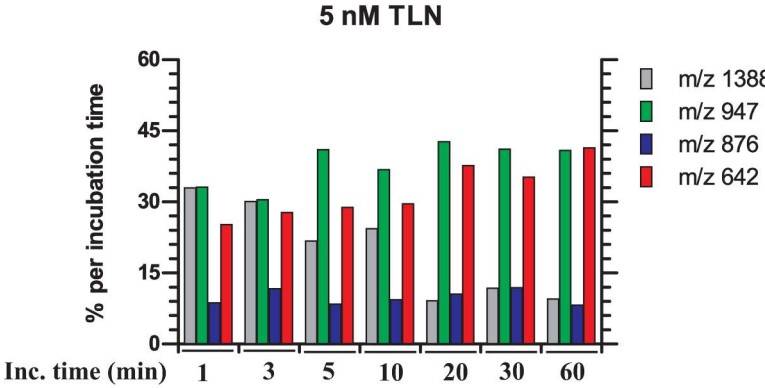

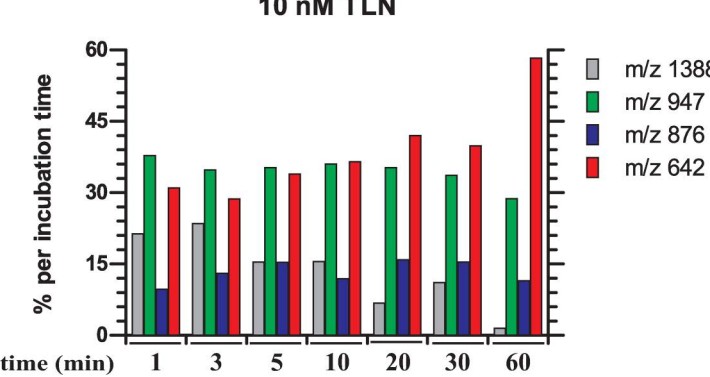

**Fig 4. MALDI-TOF MS analysis.** Relative intensities in percent of substrate Mca-Arg-Pro-Pro-Gly-Phe-Ser-Ala-Phe-Lys(Dnp)-OH (ES005) and cleavage products at 1, 3, 5, 10, 20 and 60 minutes of incubation of 50 μM ES005 with 5 and 10 nM thermolysin (TLN). The m/z value of 1388 represents the substrate ES005, whereas the m/z values of 947, 876 and 642 represent the N-terminal cleavage products after Ala-Phe, Ser-Ala, and Gly-Phe cleavage, respectively.

was not sufficient to completely inhibit 100 nM of TLN. As a result, substrate cleavage was observed immediately (0 minutes) with a pattern identical to that seen at lower TLN concentrations (S2 Fig in S1 File). After 30 minutes of incubation, the peak corresponding to full-length ES005 had disappeared. By 60 minutes, both the Ala-Phe cleavage fragment (Mca-Arg-Pro-Pro-Gly-Phe-Ser-Ala) and the Ser-Ala cleavage fragment (Mca-Arg-Pro-Pro-Gly-Phe-Ser) were no longer

detectable. The only observed fragment was the Gly-Phe cleavage product (Mca-Arg-Pro-Pro-Gly) (S2 Fig in S1 File). With increased incubation time, the peak corresponding to Mca-Arg-Pro-Pro-Gly peak (Gly-Phe cleavage) became more prominent compared to the two other product peaks. This suggests that the cleavage products Mca-Arg-Pro-Pro-Gly-Phe-Ser-Ala and Mca-Arg-Pro-Pro-Gly-Phe-Ser were further processed into Mca-Arg-Pro-Pro-Gly.

## Molecular dynamics simulations of substrate-enzyme interactions

The binding modes of ES001 with MMP-9 and MMP-14, as well as ES005 with the bacterial virulence factors, are not known from structural studies. However, the binding mode of a substrate probe molecule to a mutated mini-variant of MMP-9 has been investigated using X-ray crystallography [50]. We utilized this X-ray complex (PDB ID: **4JIJ**) as a starting point to obtain more information about the binding of ES001 to MMP-9 and MMP-14, as well as ES005 to TLN, using MD simulations. The aims were to support our MALDI-TOF MS results, examine enzyme-substrate interactions in both bacterial and human zinc proteases, and identify key amino acids constituting the most important subpockets of these enzymes.

**Interactions of ES001 with MMP-9 and MMP-14.** The only structural difference between ES001 and the substrate probe molecule co-crystallized with MMP-9 [50] is the $P_1$' residue, which is a 4-iodo-Phe in the substrate probe and Leu in ES001 (Fig 2). The X-ray crystal structure indicated that the substrate probe binds MMP-9 in an elongated conformation, with the N-terminal Mca group in the unprimed direction and the C-terminal arginine in primed direction. ES001 was docked into MMP-9 in a similar starting orientation, where the Gly occupied the $S_1$ subpocket and the Leu occupied the $S_1$' subpocket (Fig 2). During 200 ns simulations both with MMP-9 and MMP-14, the substrate maintained its orientation and elongated conformation within the binding site cleft throughout the simulations. In both enzymes, the terminal parts of ES001 showed the greatest structural flexibility, with the N-terminal Mca and the Dpa in the C-terminal being most exposed to the solvent (Fig 5). The root means square deviations (RMSDs) of the complexes from the starting complexes during MD simulations indicate structurally stable complexes, and the substrate formed stable interactions with subpockets of the enzymes (S3 Fig in S1 File). After approximately 25 ns, the RMSD for the substrate relative to the starting conformation stabilized at approximately 3 Å for MMP-9 and 4 Å for MMP-14 (S3 Fig in S1 File).

During 200 ns MD simulation with MMP-9, the backbone NH group of the Leu residue in the $P_2$ position of ES001 interacted with Ala191 (85% of frames), while the carbonyl oxygen of Gly at the $P_1$ position interacted with the catalytic zinc ion (100% of frames). The NH group of Leu at the $P_1$' position interacted with the catalytic glutamic acid (Glu402) in 30% of frames. One of the negatively charged oxygen atoms and a nitrogen atom of Dpa in $P_2$' position interacted with the catalytic zinc for most of the simulation (Fig 5). Additionally, the backbone NH group and a side chain nitrogen of Dpa formed interactions with the backbone carbonyl group of Pro421. A water molecule was bridging between Leu188 and the backbone carbonyl oxygen in $P_1$'-position, while Tyr423 interacted with the backbone carbonyl oxygen in $P_2$'-position in more than 30% of the frames (Fig 5, S4 Fig in S1 File). Overall, ES001 obtained a hydrogen bonding network with MMP-9 very similar to that of the substrate probe in the X-ray complex of MMP-9 [50].

During 200 ns MD with MMP-14, the interactions of the central regions of ES001 around the cleavages site were quite like those with MMP-9. ES001 maintained comparable interactions with the catalytic zinc ion and the catalytic glutamic acid (Glu240) as with MMP-9 (Fig 5, S4 Fig in S1 File), but the interaction fraction with Glu240 during the MD was higher for MMP-14 (Fig 5, S4 Fig in S1 File). However, the interactions around the $P_2$ and $P_2$' positions differed significantly from that in MMP-9 for most of the simulation frames (Fig 5).

**Interactions of ES005 with TLN.** MALDI-TOF MS studies showed that ES005 is cleaved at three sites (Table 2), and three MD simulations were therefore performed: 1. Gly as the $P_1$ residue and Phe as the $P_1$' residue (GF-MD). 2. Ala as the $P_1$ residue and Phe as the $P_1$' residue (AF-MD). 3. Ser as the $P_1$ residue and Ala as the $P_1$' residue (SA-MD). ES005 is larger than both ES001 and the substrate probe co-crystallized with MMP-9 [50]. As a result, docking ES005 into the catalytic site of TLN was more challenging than for ES001, making the starting complexes for MDs less certain compared to those of ES001.

## MMP-9/ES001

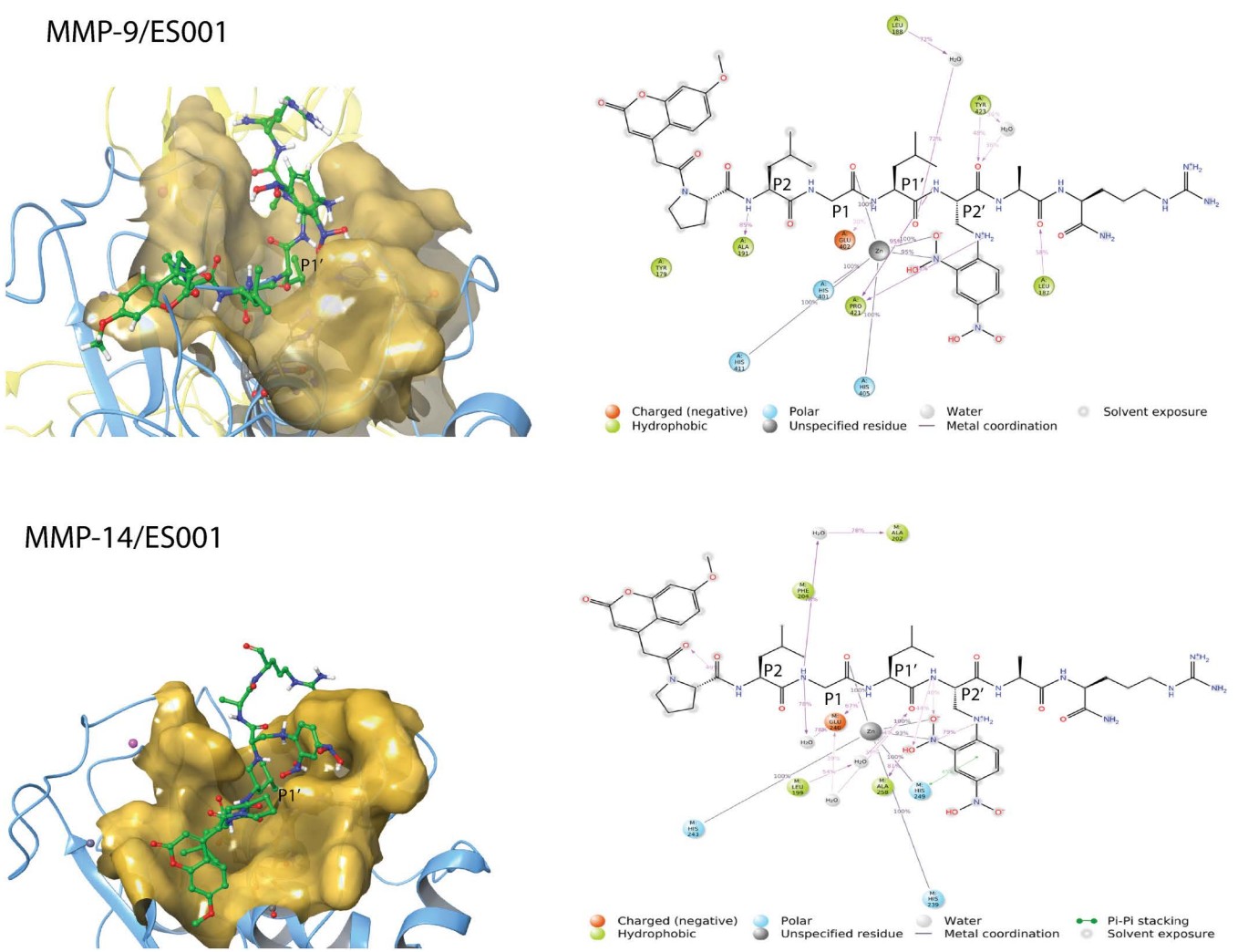

## MMP-14/ES001

**Fig 5. Molecular interactions of ES001 with MMP-9 and MMP-14.** Left: Close up view of the binding cleft of MMP-9 and MMP-14 with ES001 (green) at the catalytic site. Oxygen atoms of ES001 in red, nitrogen atoms in dark blue, while hydrogen atoms connected to nitrogen or oxygen atoms are in white. The surface of the binding cleft depicted in yellow, with the $P_1$' residue indicated. The Cα trace of the catalytic part is shown in light blue. MD frames after 161 ns of simulation with MMP-9 and after 125 ns with MMP-14 are used as representatives for the MD frames. The yellow backbone Cα trace of MMP-9 highlights the fibronectin repeats. Right: 2D illustration of the complexes with the most important ES001-enzyme interactions, where amino acids within 4 Å of the substrate are included. The percentages represent the frequency of these interactions observed in MD frames, with only those present in more than 30% of the frames included.

During the GF-MD, ES005 retained an elongated conformation throughout the entire simulation, with the N-terminal Mca extending in the unprimed direction and the C-terminal Dnp in the primed direction (Fig 6). This orientation was like that observed for ES001 in MMP-9 and MMP-14 (Fig 5) and in the X-ray structure of the substrate probe with MMP-9 [50]. The RMSD of ES005 relative to its starting conformation remained stable around 6 Å from the beginning of the 200 ns sampling period (S5 Fig in S1 File), suggesting both internal structural stability and consistent interactions with TLN throughout the simulation. The catalytic zinc interacted with the backbone carbonyl group of both Gly in the $P_1$ position and Pro in the $P_2$ position. Asn112 interacted with the backbone NH group of both Phe in $P_1$' and Ser in $P_2$', while Arg203 interacted with the backbone CO in $P_1$' position. Asn111 interacted with side chain of Ser in $P_2$', while His142 was stacking with the side chain Phe in $P_1$' position (Fig 6).

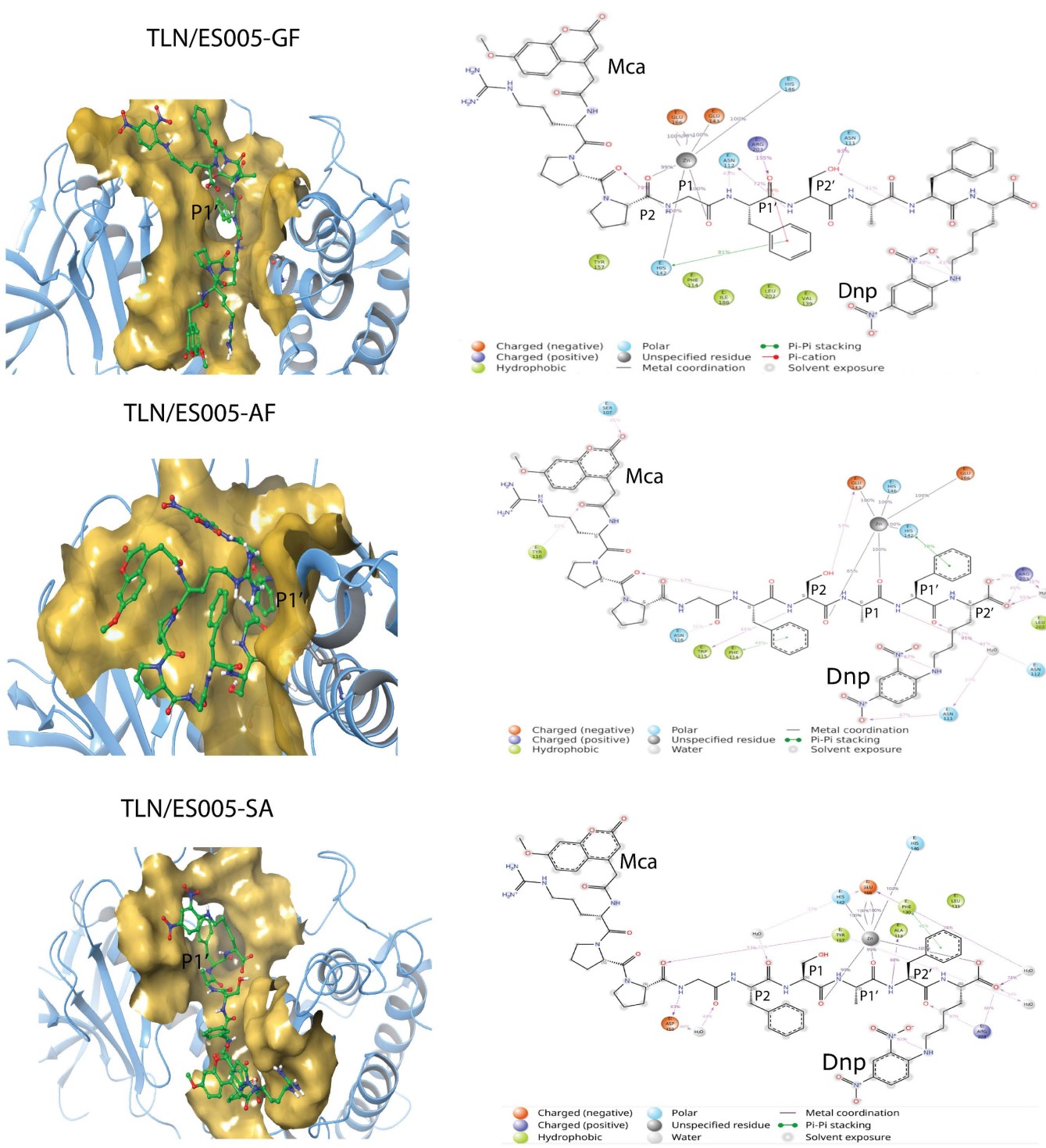

**Fig 6. Molecular interactions of ES005 with TLN.** Left: Close ups of the TLN binding cleft surface (yellow) during the three MD simulations: TLN/ES005-GF, TLN/ES005-AF and TLN/ES005-SA. The Cα-trace of TLN is shown in light blue. ES005 atoms are colour coded: carbon in green, oxygen in red, and nitrogen in dark blue. The representative coordinate sets were selected at 137 ns, 160 ns, and 200 ns for the TLN/ES005-GF, TLN/ES005-AF, and TLN/ES005-SA simulations, respectively. Right: 2D illustration of the TLN/ES005 complexes, highlighting the most significant interactions. Amino acids within 4 Å of the substrate are included, with interaction frequencies indicated as percentage of MD frames. Only interaction present in more than 30% of the frames are included.

During the AF- MD, the RMSD of ES005 relative to its starting conformation stabilized around 7 Å (S5 Fig in S1 File). After approximately 100 ns, ES005 obtained a very stable conformation and stayed in that conformation for the rest of the simulation. In the first 100 ns, the N-terminal region of ES005 did not have binding partners within the enzyme, and large portions of the substrate (especially in the N-terminal parts) were highly exposed to solvent. However, after 100 ns, the N-terminal Mca group folded back towards the C-terminal Dnp group, resulting in a folded and very stable conformation of the substrate. Notably, during the AF-MD, the unprimed amino acids of ES005 were more important for forming interactions with amino acids in TLN than in the GF-MD complex (Fig 6). The backbone carbonyl group of Gly ($P_4$ position) obtained hydrogen bonds with Asn116, while the backbone carbonyl of Pro in $P_6$ position formed an internal hydrogen bond with the amide group of Phe in $P_3$ position. The backbone NH of Ser in $P_2$ interacted with Trp115. The side chain of Phe in $P_3$ was stacking with Phe114. The backbone carbonyls of Ser ($P_2$) and Ala ($P_1$) interacted with the catalytic zinc ion. The backbone NH of Phe ($P_1'$) interacted with Asn112, while its side chain stacked with His142. Arg203 interacted with at the C-terminal end of ES005 (Fig 6).

The RMSD plots during the SA-MD indicate that the conformation of ES005 was more unstable relative to the starting conformation than in the AF-MD and GF-MD. The RMSD (S5 Fig in S1 File) varies between approximately 3.4 Å and 6.8 Å, suggesting considerable internal structural fluctuations. Frame analysis revealed that the largest fluctuations occurred in the N-terminal unprimed region of the substrate. During the SA-MD, the carbonyl group of the amino acid in $P_1$ position interacted with the catalytic zinc (Fig 6), similarly to the AF- and GF-MD simulations. Additionally, the carbonyl group of the Ala in the $P_1'$ position and a negatively charged oxygen atom at the C-terminus also interacted with the catalytic zinc (Fig 6). Arg203 further stabilized the C-terminal region by interacting with the carbonyl group of Phe in the $P_2'$ position and a terminal oxygen of ES005 (Fig 6).

During catalytic cleavage, the incoming substrate is presumed to displace a water molecule coordinated to the catalytic zinc. This displacement allows the general base (Glu143 of the HEXXH motif) to accept a proton from the zinc polarized water molecule and transfer it to the leaving nitrogen of the scissile bond [58,59]. Throughout the cleavage process, the side chain oxygen of Asn112 in TLN and the backbone carbonyl of Ala113 are suggested to form hydrogen bonds with the protonated nitrogen in the $P_1'$ position of the scissile bond. The residual interaction fractions were monitored during MDs. The residual interaction fraction categorizes the interactions (contacts) between a ligand and a particular residue (amino acid) by type and summarises the interactions. One protein residue may have several contacts for the same type of interaction, but also several types of contacts with the ligand during MD. Therefore, it is possible with values larger than 1.0 for the residual interaction fraction. In both the AF-MD and GF-MD, Asn112 formed hydrogen bonds with the backbone NH group of the $P_1'$ residue, while the general base Glu143 remained close to the NH group of the $P_1'$ residue. As indicated by the residual interaction fraction, Asn112 played a crucial role in interacting with ES005 in both simulations, with interaction fractions of approximately 1.5 during AF-MD and 2.4 during GF-MD (S6 Fig in S1 File). In contrast, during the SA-MD, Asn112 did not interact with the NH group of the $P_1'$ residue (Fig 6) and had only limited involvement in ES005 interactions, with an interaction fraction of approximately 0.2 (S6 Fig in S1 File). The residual interaction fractions show that 13 hydrogen bonds were present between TLN and ES005 during the AF-MD, compared to 10 in the GF-MD and 12 in the SA-MD. However, the residual interaction fractions were much higher for the AF- and GF-MDs than for the SA-MD (S6 Fig in S1 File). Similarly, the fractions for ionic and hydrophobic interactions were generally higher in the AF- and GF-MDs, while the number of water bridges was relatively consistent across all three complexes.

The three MD simulations showed that ES005 fits better into the catalytic site and obtain more favourable interactions with TLN when the Ala or Gly occupy the $S_1$ subpocket and Phe the $S_1'$ subpocket, compared to when Ser occupies the $S_1$ and Ala the $S_1'$ subpocket. This aligns with the MALDI-TOF MS results. The SA-MD suggests that the side chain of Ala is too short to obtain a maximum of favourable contacts with amino acids within the $S_1'$ subpocket. This finding is consistent with previous studies indicating that TLN preferentially cleaves at the N-terminal side of hydrophobic or bulky amino side chains, such as Leu, Phe, Ile, and Val. However, TLN may also cleave at the N-terminal side of residues like Met, His, Tyr, Ala, Asn, Ser, Thr, Gly, Lys, Glu, or Asp [3,60].

**Subpocket amino acids.** The amino acids within the subpockets during MD simulations of ES001 with MMP-9 and MMP-14 and ES005 with TLN (GF-MD) are listed in Table 3. Amino acids in corresponding subpockets of PLN and ALN were identified through sequence alignments of the catalytic domains of TLN, PLN, and ALN (S7 Fig in S1 File). The table is based on MD simulations with substrates that obtain favourable interactions within the different subpockets for cleavage. However, inhibitors may occupy other regions and amino acids of a subpocket than substrates. As a result, Table 3 differs from a corresponding table based on docking of series of bisphosphonate- and catechol-containing inhibitors by Rahman et al. 2021 [35]. The largest difference between the present and the previous table is in the number of amino acids in the $S_1$' subpocket that interacted with substrates in the present study and with the inhibitors in the previous study.

To be potential drug candidates for treating of *P. aeruginosa* or *S. aureus* infections, PLN and ALN inhibitors should not inhibit human zinc proteases. The amino acids forming the substrate subpockets are quite similar between the bacterial and human zinc proteases studied (Table 3). The major subpocket for substrate cleavage specificity is the $S_1$' subpocket, which is mainly hydrophobic in both bacterial and human enzymes. However, key differences in this subpocket may be important for designing inhibitors that selectively target the virulence factors over MMP-9 and MMP-14. The $S_1$' subpocket is both deeper and wider in the human enzymes, suggesting they can accommodate larger side chains compared to the bacterial enzymes. Notably, a positively charged Arg is present in the $S_1$' subpocket of all three bacterial enzymes (S7 Fig in S1 File), while this residue is absent in MMP-9 and MMP-14 (Table 3). Although MMP-9 contains an Arg in the $S_1$' subpocket, it is located much deeper in the pocket than in the bacterial enzymes. Targeting the Arg in the bacterial enzymes could be a promising strategy for developing inhibitors with stronger binding affinity for bacterial virulence factors than for MMP-9 and MMP-14.

## Enzyme inhibition and docking of the phosphinic compounds

**$K_m$ values of fluorescence quenched substrates.** Under the experimental conditions used in the present work (1% DMSO in all assays), the $K_m$ value of ES001 with recombinant MMP-9(MT) was 2.7±0.5 μM (n=3). The $K_m$ values for ES001 with MMP-9(A) and MMP-14, as well as for ES005 with TLN, PLN, and ALN, were previously determined under the same conditions [35]. To summarize, the $K_m$ values for ES001 with MMP-9(A) and MMP-14 were 4±1 μM and 4.9±0.4 μM, respectively. For ES005, the $K_m$ values with ALN, PLN, and TLN were 76±7 μM, 24±8 μM, and 6±1 μM, respectively. Although the estimated $K_m$ values for ALN and PLN were regarded less certain due to substrate quenching at concentration above 10 μM, the values were consistent with previous measurements conducted either without DMSO [38] or with 5% DMSO [37].

**Quenching experiments.** All compounds were tested for possible quenching of the fluorescence product. Experiments were performed with varying concentrations of the putative inhibitors (0–100 μM) against varying concentrations of the fluorescence product (McaPLG-OH) of ES001, as previously described for procaspase-activating compound 1(PAC-1)

**Table 3. Amino acids within different subpockets of MMP-9, MMP-14, TLN, PLN and ALN.** Amino acids included are those located near substrate side chains during MD simulations with MMP-9, MMP-14, and TLN (GF-MD). Subpocket amino acids of PLN and TLN were identified based on amino acid sequence alignments with TLN (S7 Fig in S1 File).

| Subpocket | MMP-9 | MMP-14 | TLN | PLN | ALN |
|---|---|---|---|---|---|
| $S_1$ | H190, A191, E402, H405 | F198, A200, A202, E240, H243 | A113, F114, Y157 | A113, Y114, Y155 | A116, A117, Y160 |
| $S_1$' | Y393, S394, F396, L397, V398, H401, L418, Y420, P421, M422, Y423, R424, T426 | L199, L235, N231, V236, V238, H239, F260, Y261, Q262, F263 | N112, M120, S134, D138, V139, H142, G189, L202, R203 | N112, M120, L132, D136, V137, H140, G187, L197, R198 | N115, M123, S137, D141, V142, H145, G187, L200, R201 |
| $S_2$ | A191, P193 | H201, Y203, F204 | W115, E143, H146 | W115, E141, H144 | W118, E146, H149 |
| $S_2$' | G186, L187, Y218, H411 | H249, S250, S251, P259 | N111, F130, L133, Y193 | E111, Y130, L132, K191 | N114, F133, L136, Y191 |

and isatin derivatives [37], and outlined in Materials and Methods. These experiments revealed that none of the phosphinic compounds quenched the fluorescence product. However, some compounds showed background fluorescence at the emission and excitation wavelengths used, though this did not interfere with the inhibitory assays, as the enzymatic reactions were followed continuously. These findings agree with our previous results for bisphosponate and catechol-containing compounds [35], as well as for dipicolylamine (DPA), tripicolylamine (TPA), tris pyridine ethylene diamine (TPED), and various pyridine and thiophene deriviaties [40]. In contrast, PAC-1 and isatin derivatives were found to significantly quench the fluorescence product [37].

**Inhibitory effects and binding modes of the phosphinic compounds.** Except for H-3 against MMP-14, the compounds were not slow binders to any of the proteases (S8 Fig in S1 File). For H-3, inhibition of MMP-14 appeared to stabilize after a 15 min pre-incubation. Among the tested compounds, only H-2 showed strong inhibition across all five proteases, with the most potent effects observed against MMP-9 and MMP-14 (Fig 7). In contrast, H-1 exhibited strong inhibition exclusively against the two human proteases, while H-3, H-4, and H-5 showed weak or no inhibition of the enzymes. $K_i$ values were determined for the enzymes where 100 µM of a phosphinic compound resulted in ≥ 50% inhibition (Table 4). H-2 exhibited the strongest binding to the MMP-9 variants (MMP-9(MT) and MMP-9(A)), with $K_i$ values

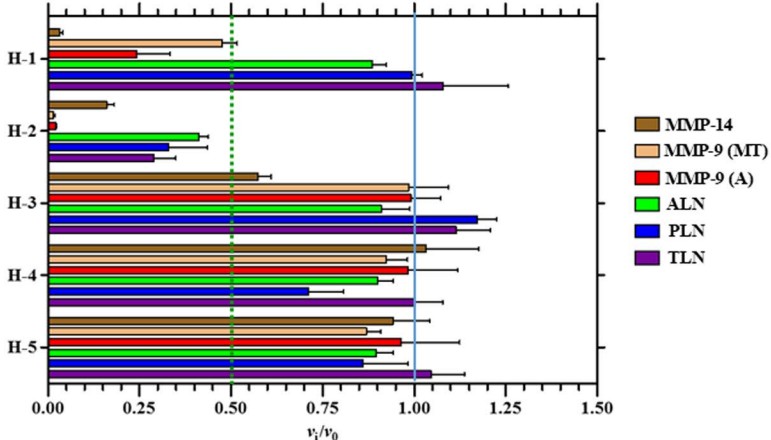

**Fig 7. Inhibitory effects.** The inhibitory effects of 100 µM of the H-compounds on the enzymatic activity of MMP-14, MMP-9(MT), MMP-9(A), TLN, PLN and ALN. All compounds showed fast inhibitory binding against all enzymes, except for H-3 against MMP-14. For the fast inhibitory binding, the $v_i/v_0$ (mean±s.d.) was based on the mean values of 0-30 min pre-incubation. For H-3 against MMP-14, the $v_i/v_0$ (mean±s.d.) was based on the mean values of 15-30 min pre-incubation, i.e., where the time curve in S8 Fig in S1 File flattens out.

**Table 4.** $K_i$ **values of the phosphinic compounds H-1 and H-2 for MMP-14, MMP-9(A), MMP-9(MT), ALN, PLN and TLN. The $K_i$±s.d. values were determined for compounds that reduced enzymatic activity by 50% or more at a concentration 100 µM. The s.d. of the $K_i$ values accounts for the combined s.d. from both the $IC_{50}$ and $K_m$ values. The contribution of the $K_m$ s.d. is weighted relative to the [S]/$K_m$ ratio in the calculation of $K_i$ from $IC_{50}$ (Eqn. 3).**

| Compound | $K_i$±s.d. (µM) | | | | | |
|---|---|---|---|---|---|---|
| | ES001 | | | ES005 | | |
| | MMP-14 | MMP-9(A) | MMP-9(MT) | ALN | PLN | TLN |
| H-1 | 0.89±0.04 | 19±3 | 30±4 | n.d.[a] | n.d. | n.d. |
| H-2 | 3.0±0.1 | 0.56±0.07 | 0.53±0.06 | 80±4 | 40±2 | 2.5±0.2 |

[a]*n.d., not determined*

around 0.5 µM. The compound bound with approximately equal affinity to MMP-14 and TLN, although with lower affinity than for MMP-9. A much weaker binding was observed for PLN and ALN, although the binding was stronger to the former enzyme.

H-2 was docked into the active site of all five MPs using induced fit docking. The highest scoring pose of H-2 with MMP-9 (gold-yellow in Fig 8) had a docking score of −16.3 kcal/mol. All enantiomers were initially docked, but the RSS enantiomer (Fig 8) had the highest score and was also used in the enzyme inhibition kinetics studies. The X-ray structure of the SSS enantiomer bound to MMP-11 is available (PDB ID: **1HV5**) [42], and this complex was superimposed with the MMP-9/H-2 induced fit docking complex in Fig 8A. The structures showed large similarities: in both, H-2 interacts with both primed and unprimed subpockets, and the propylphenyl group occupies the $S_1'$ subpocket and penetrates deeply into the pocket. The most striking difference was that the indole ring occupied the $S_2'$ subpocket in the X-ray complex (SSS enantiomer), while in the best docked complex (RSS enantiomer), the ring extended more towards the $S_3'$ subpocket (Fig 8C). Additionally, the positions of the other phenyl rings also differed between the two complexes.

The position of H-2 (RSS) in MMP-14 (Fig 9) was like its position in MMP-9 (Fig 8). However, in the highest scoring pose, the indole ring oriented towards the $S_2'$ subpocket (Fig 9), resembling the X-ray complex of the SSS-enantiomer with MMP-11 (Fig 8). The docking score for H-2 in MMP-14 was − 14.9 kcal/mol.

In agreement with the obtained $K_i$ values (Table 4), the induced fit docking studies indicated that H-2 binds weaker to TLN, PLN, and ALN than to MMP-9 and MMP-14. The docking scores for H-2 were as follows: PLN: −13.0 kcal/mol, ALN: −13.0 kcal/mol, TLN: −13.4 kcal/mol. The main differences between the bacterial MPs and human MMPs was the binding orientation of H-2. In the bacterial MPs (Figs 10–12), the phenyl ring closest to the phosphinyl group entered the $S_1'$ subpocket, while in MMP-9 (Fig 8) and MMP-14 (Fig 9), the propylphenyl group occupied this pocket. In the bacterial enzymes, the propylphenyl group extended towards the unprimed sites, suggesting that the $S_1'$ subpocket of the virulence factors is too small to accommodate the longer propyl chain. As a result, the shorter phenyl group entered the $S_1'$ subpocket instead. This aligns with previous studies on hydroxamic acid-based compounds as ZBG, which showed that the $S_1'$ subpockets of MMPs can occupy larger groups compared to TLN, PLN, and ALN [36,38,39]. Notably, H-2 interacted with both prime and unprimed subsites in the bacterial MPs, consistent with its docking behaviour in MMP-9 and MMP-14. These findings indicate that H-2 and other phosphinyl compounds may mimic substrate transition states while interacting with bacterial zinc MPs and human MMPs.

H-1 binds stronger to MMP-14 than to the two variants of MMP-9, with $K_i$ values of 0.89 µM, 19 µM and 30 µM, respectively (Table 4). Induced fit docking of H-1 into MMP-9 and MMP-14 was performed with both the RSS and SSS enantiomer, which differ at the chiral centre closest to the phosphinyl group (Fig 1). The X-ray structure of the SSS enantiomer in complex with MMP-12 is available (PDB ID: **5CZM**) [31], showing that the large isoxazolyl side chain of H-1 (Fig 1) penetrates deeply into the $S_1'$ subpocket of MMP-12. Docking studies revealed that this side chain similarly enters the $S_1'$ subpocket of MMP-9 (Fig 13) and MMP-14 (Fig 14). Both the SSS and RSS enantiomer could be docked into the active site of MMP-9 and MMP-14, but the SSS enantiomer scored better in both cases, with docking scores of −13.2 kcal/mol for MMP-9 and −9.2 kcal/mol for MMP-14. In contrast, the scoring of the RSS enantiomer was-7.2 kcal/mol for MMP-9 and −6.3 kcal/mol for MMP-14. These docking scores indicated that H-1 binds stronger to MMP-9 than MMP-14, which contradicts the experimentally determined $K_i$ values (Table 4). In the bacterial enzymes, the isoxazolyl side chain appeared too bulky to fit into the $S_1'$ subpocket, and no clear interactions with any of the enzyme subpockets were observed. Most likely, this explains why H-1 failed to bind PLN, TLN, and ALN.

Previously, H-1 (RXP470) (Fig 1) was tested as an inhibitor of various metalloproteases (MMP-1,-2,-3,-7,-8,-9,-10,-11,-12,-13,-14), angiotensin-converting enzyme (ACE), neprilysin (NEP), and tumour necrosis factor-α-converting enzyme (ADAM17 or TACE) [41,61]. Consistent with our results, they found that H-1 binds stronger to MMP-14 than MMP-9, with reported $K_i$ values of 0.14 and 1.3 µM, respectively [41]. Notably, these previously obtained $K_i$ values were lower than those obtained in the present study (Table 4). H-2 (RXPO$_3$) was also tested against various matrix metalloproteases

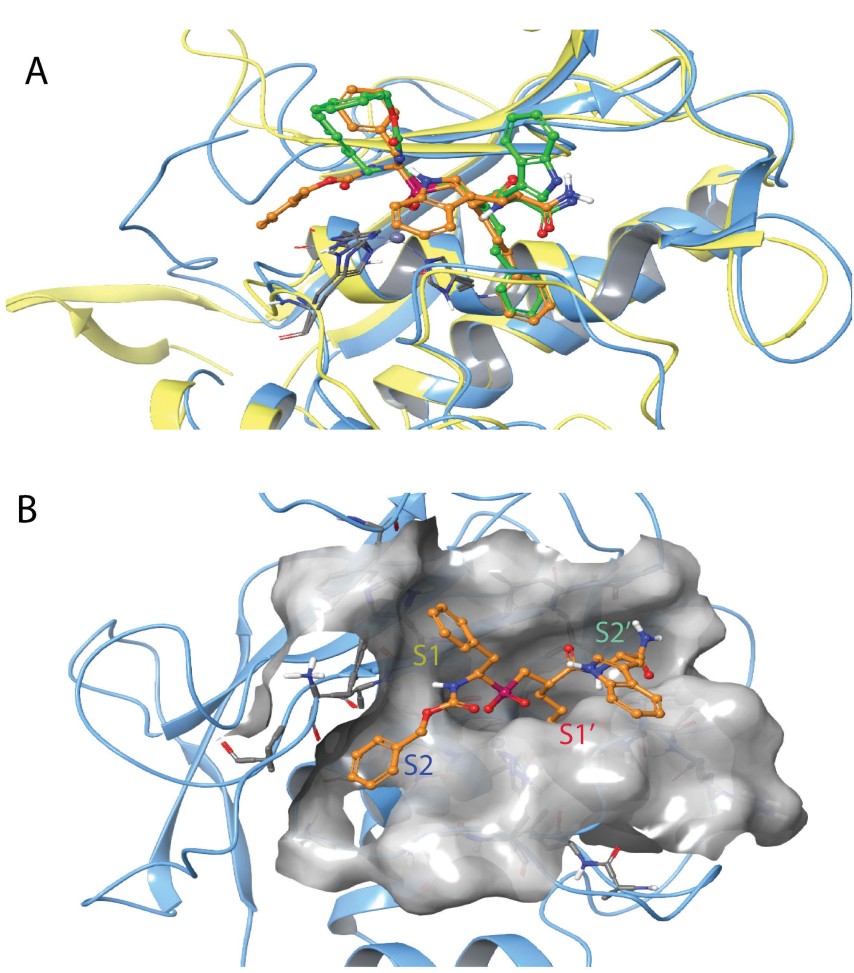

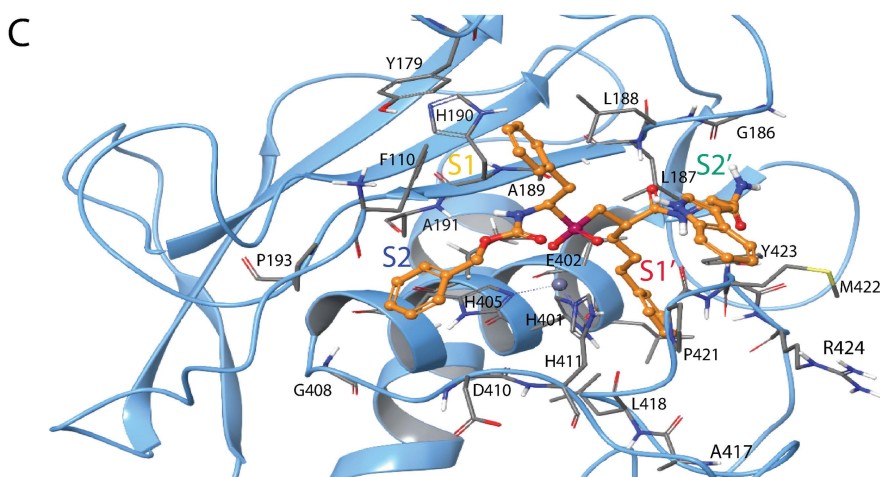

**Fig 8. Induced fit docking of H-2 with MMP-9. A:** The docked complex of H-2 (gold-yellow, RSS enantiomer) with MMP-9 (light blue Cα-trace) super-imposed with the X-ray structure of the MMP-11 complex (PDB ID: **1HV5**) (yellow Cα-trace) with H-2 (green, SSS enantiomer). H-2 oxygen atoms are shown in red, nitrogen atoms in dark blue, the catalytic zinc in grey and the phosphate group in dark red. The RSS enantiomer was used in the enzyme

kinetics studies. **B.** The same docked H-2-MMP-9 complex as in A with identical colour coding. The surface of the binding cleft is displayed in grey. **C:** The same H-2-MMP-9 complex as in A and B, with identical colour coding and approximately in the same view as in **B**. The most important amino acids involved in binding are included. The S$_1$, S$_1$', S$_2$, and S$_2$' subpockets are indicated in B and C. Amino acids within different subpockets of MMP-9 are shown in Table 3. 2D illustration of the interactions between H-2 and MMP-9 is shown in S9 Fig in S1 File.

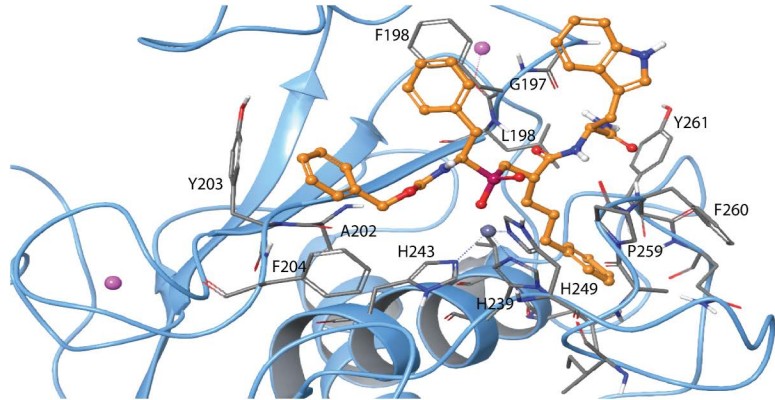

**Fig 9. Induced fit docking of H-2 with MMP-14.** The docked complex of H-2 (RSS enantiomer) with MMP-14. Colour coding as in Fig 8, and in addition, two calcium ions are shown in pink. Amino acids in different subpockets of MMP-14 are shown in Table 3. The complex is seen in same view as the H2-MMP-9 complex in Fig 8. 2D illustration of the interactions between H-2 and MMP-14 is shown in S10 Fig in S1 File.

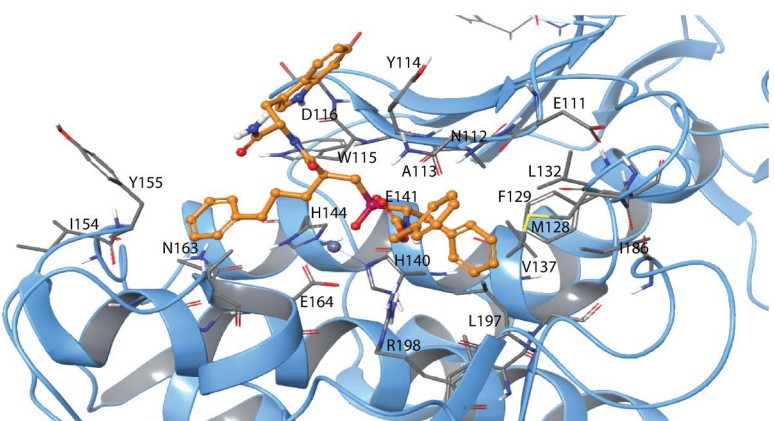

**Fig 10. Induced fit docking of H-2 with PLN.** The docked complex of H-2 (RSS enantiomer) with PLN. Colour coding as in Fig 8. Amino acids in different subpockets of PLN are shown in Table 3. The complex is seen in same view as the H2-MMP-9 complex in Fig 8. 2D illustration of the interactions between H-2 and PLN is shown in S11 Fig in S1 File.

(MMP-1,-2,-7,-8,-9,-11,-14) [34]. In agreement with our findings, H-2 bound stronger to MMP-9 than to MMP-14, with $K_i$ values of 10 nM and 40 nM, respectively. Another study showed that that the RSS enantiomer of H-2 bound stronger to MMP-14 than the RRS enantiomer, with $K_i$ values of 90 and 386 nM, respectively [33]. In both studies, the reported $K_i$ values were lower than those obtained in the present study for the RSS enantiomer of H-2. Differences in the experimental setup between the current and previous studies of H-1 and H-2 [33,34,41] may explain these discrepancies. Factors such

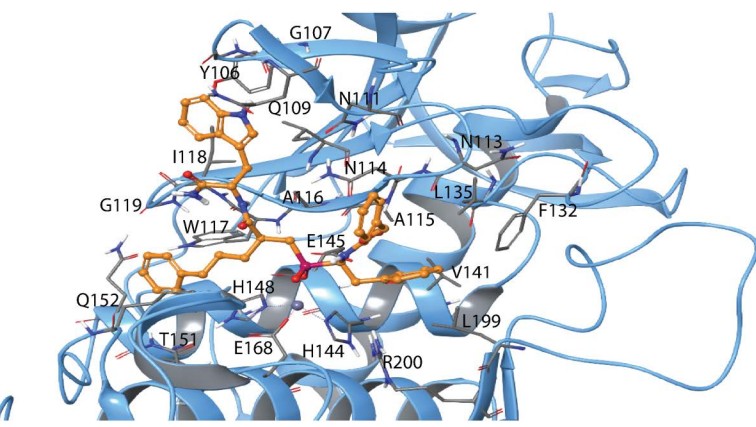

**Fig 11. Induced fit docking of H-2 with ALN.** The docked complex of H-2 (RSS enantiomer) with ALN. Colour coding as in Fig 8. Amino acids in different subpockets of ALN are shown in Table 3. The complex is seen in same view as the H2-MMP-9 complex in Fig 8. 2D illustration of the interactions between H-2 and ALN is shown in S12 Fig in S1 File.

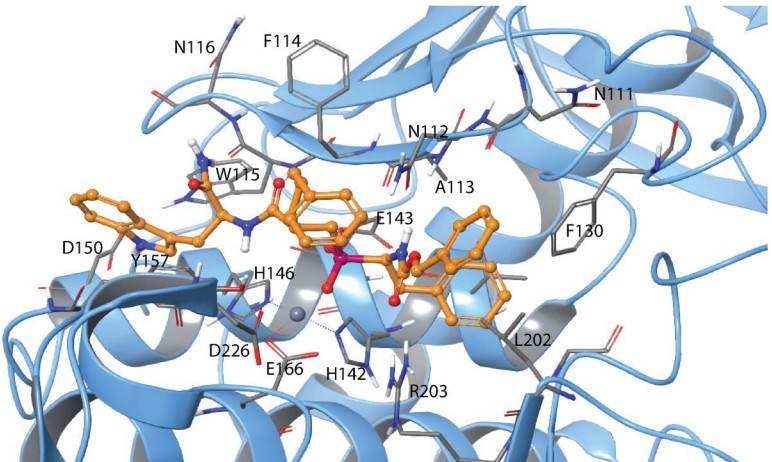

**Fig 12. Induced fit docking of H-2 with TLN.** The docked complex of H-2 (RSS enantiomer) with TLN. Colour coding as in Fig 8. Amino acids in different subpockets of TLN are shown in Table 3. The complex is seen in same view as the H2-MMP-9 complex in Fig 8. 2D illustration of the interactions between H-2 and TLN is shown in S13 Fig in S1 File.

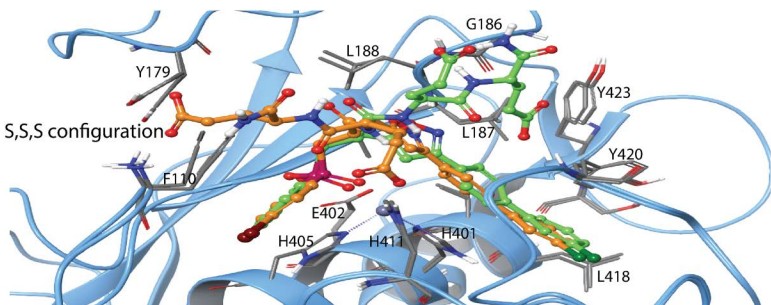

**Fig 13. Induced fit docking of H-1 with MMP-9.** The docked complex of H-1 (yellow SSS enantiomer, green RSS enantiomer) with MMP-9. Colour coding of the enzyme as in Fig 8, with H-1 atom colours matching the scheme used for H-2 in the same figure. 2D illustration of the interactions between H-1 and MMP-9 is shown in S14 Fig in S1 File.

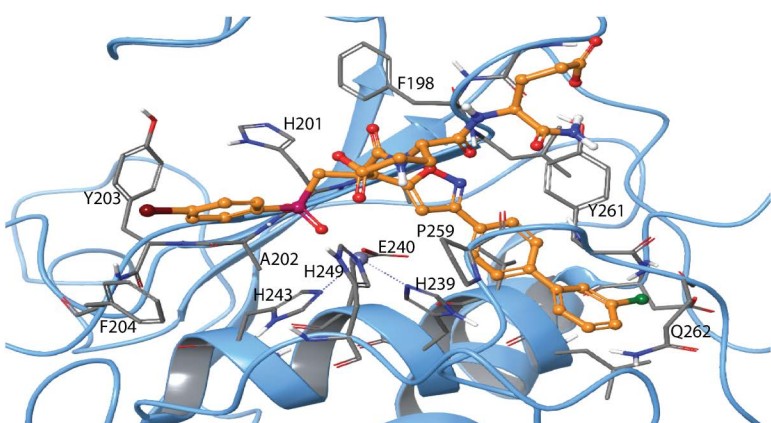

**Fig 14. Induced fit docking of H-1 with MMP-14.** The SSS enantiomer of H-1 docked into MMP-14. The enzyme is colour coded as in Fig 8, With H-1 atom colours matching those of H-2 in the same Figure. 2D illustration of the interactions between H-1 and MMP-14 is shown in S15 Fig in S1 File.

as buffer compositions, pH and assay reaction temperature likely influenced the results. Previous studies showed that the binding of H-1 (RXP470) to MMP-12 is pH dependent, with the $K_i$ value increasing approximately ten times for each unit increase in buffer pH [31,32]. Additionally, assay temperature can impact binding affinity, as seen in *Drosophila lebanonensis* alcohol dehydrogenase (ADH), where the dissociation constant ($K_{EO,I}$) of pyrazole from the enzyme-NAD$^+$-pyrazole complex increased with rising temperature in the pH range of 7–10 [62]. In the present study, both the buffer pH and assay temperature were higher than in previous studies, which may at least partially explain the higher $K_i$ values observed for H-1 and H-2 with MMP-9 and MMP-14.

The phosphinic di- and tri-peptide compounds H-3 to H-5, featuring either a carboxylic acid or a carboxylic ethyl ester at the C-terminal $P_1$' position (Fig 1), were previously studied as potential carboxy-exopeptidase inhibitors. H-4, for instance, was among several phosphinic di- and tri-peptide compounds designed and tested against angiotensin-converting enzyme 2 (ACE2), carboxypeptidase A, and ACE [44]. Given their design and prior evaluation, it was unsurprising that H-3 to H-5 exhibited weak or no inhibition of the five proteases investigated in the present work (Fig 7).

## Conclusions

The MALDI-TOF MS results showed that the substrate ES005 (Mca-Arg-Pro-Pro-Gly-Phe-Ser-Ala-Phe-Lys(Dnp)-OH) was cleaved by TLN at three distinct sites: Ala-Phe, Gly-Phe, and Ser-Ala. The fraction of cleavage products at short incubation times followed the order: Ala-Phe > Gly-Phe >> Ser-Ala. Based on these results, three ES005-TLN complexes were constructed through molecular modelling, with the cleavage site amino acids within the $S_1$- and $S_1$'-subpockets of TLN. MD simulations revealed that the structure of ES005 remained significantly more stable relative to TLN when the Ala-Phe and Gly-Phe pairs occupied the $S_1$- and $S_1$' subpockets, compared to the Ser-Ala pair. The interaction fractions of amino acids participating in non-bonded interactions with ES005 were also much higher for the Ala-Phe and Gly-Phe pairs than for Ser-Ala. These findings agree with the previously suggested substrate specificity of TLN.

Compound H-1 inhibited MMP-14 and two variants of MMP-9, with inhibition constants ranging from 0.89 to 30 µM, but showed no inhibition of the bacterial enzymes. Induced fit docking indicated that the bulky isoxazyl side chain was too large to fit the $S_1$' subpocket of the bacterial enzymes, and no clear interactions with the enzyme subpockets were observed, which likely explains the lack of binding to PLN, TLN and ALN. In contrast, compound H-2 inhibited the human MMPs with inhibition constants ranging from 0.53 µM (MMP-9) to 3.0 µM (MMP-14), and the bacterial enzymes with inhibition constants ranging from 2.5 µM (TLN) to 80 µM (ALN). Induced fit docking indicated that the main difference in binding

mode was that, in the bacterial MPs, the phenyl ring closest to the phosphinyl group of compound H-2 entered the $S_1$' subpocket, whereas in the human MMPs, the larger propylphenyl group occupied this pocket.

The induced fit docking studies indicated that the isoxazyl side chain of H-1 was too big to enter the $S_1$' subpocket of the bacterial virulence factors. The present study, and our previous studies [35,36,39,40] have indicate that the size and shape of the $S_1$' subpocket is an important determinant for selectivity between the studied bacterial virulence factors and human MMPs. In the present study, only H-2 inhibited the bacterial virulence factors and bound both unprimed and primed subpockets of the virulence factors. Table 3 shows that there are structural differences between the bacterial virulence factors and the human MMPs in these subpockets. Designing inhibitors with a chemical group that favourably fits into the $S_1$'subpocket, and in addition make use of the differences in the unprimed subpockets between the bacterial virulence factors and human MMPs may increase the selectivity for the bacterial virulence factors compared with human MMPs.

## Supporting information

**S1 File.  A pdf file containing the supporting Table S1 and supporting figs (S1 Fig – S15 Fig).**
(PDF)

**S2 File.  IC$_{50}$ calculations of H-1 against MMP-9A and MMP-9T.** An Excel file with enzyme kinetic raw data.
(XLSX)

**S3 File.  IC$_{50}$ calculations of H-1 against MMP-14.** An Excel file with enzyme kinetic raw data.
(XLSX)

**S4 File.  IC$_{50}$ calculations of H-2 against MMP-9.** An Excel file with enzyme kinetic raw data.
(XLSX)

**S5 File.  IC$_{50}$ calculations of H-2 against TLN.** An Excel file with enzyme kinetic raw data.
(XLSX)

**S6 File.  Time dependent inhibition.** An Excel file with enzyme kinetic raw data.
(XLSX)

**S7 File.  Curves giving IC$_{50}$- and Ki-values of H-1 and H-2.** A PowerPoint file with enzyme kinetic data.
(PPTX)

**S8 File.  Quenching results for the compounds.** A PowerPoint file with enzyme kinetic data.
(PPTX)

## Acknowledgments

The phosphinic compounds (Fig 1) were a kind gift from professor Athanasio Yiotakis, University of Athens, Greece.

## Author contributions

**Conceptualization:** Ingebrigt Sylte, Tor Haug, Klara Stensvåg, Jan-Olof Winberg.

**Data curation:** Fatema Amatur Rahman, Ida Kristine Østnes Hansen, Imin Wushur, Bibek Chaulagain, Tra-Mi Nguyen, Olayiwola Adedotun Adekoya, Nabin Malla.

**Formal analysis:** Fatema Amatur Rahman, Ida Kristine Østnes Hansen, Imin Wushur, Bibek Chaulagain, Tra-Mi Nguyen, Olayiwola Adedotun Adekoya, Nabin Malla.

**Funding acquisition:** Ingebrigt Sylte, Tor Haug, Klara Stensvåg, Jan-Olof Winberg.

**Investigation:** Ingebrigt Sylte, Fatema Amatur Rahman, Imin Wushur, Tor Haug, Klara Stensvåg, Bibek Chaulagain, Tra-Mi Nguyen, Olayiwola Adedotun Adekoya, Jan-Olof Winberg.

**Methodology:** Ingebrigt Sylte, Fatema Amatur Rahman, Ida Kristine Østnes Hansen, Imin Wushur, Tor Haug, Klara Stensvåg, Jan-Olof Winberg.

**Project administration:** Ingebrigt Sylte, Tor Haug, Klara Stensvåg, Jan-Olof Winberg.

**Resources:** Ingebrigt Sylte, Tor Haug, Klara Stensvåg, Jan-Olof Winberg.

**Software:** Imin Wushur, Bibek Chaulagain.

**Supervision:** Ingebrigt Sylte, Tor Haug, Klara Stensvåg, Jan-Olof Winberg.

**Validation:** Fatema Amatur Rahman, Ida Kristine Østnes Hansen, Imin Wushur.

**Visualization:** Ingebrigt Sylte, Fatema Amatur Rahman, Ida Kristine Østnes Hansen, Imin Wushur, Tor Haug, Jan-Olof Winberg.

**Writing – original draft:** Ingebrigt Sylte, Fatema Amatur Rahman, Ida Kristine Østnes Hansen, Imin Wushur, Jan-Olof Winberg.

**Writing – review & editing:** Ingebrigt Sylte, Ida Kristine Østnes Hansen, Tor Haug, Klara Stensvåg, Jan-Olof Winberg.

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
