## [Decision Letter · Decision Letter 0]

PONE-D-25-23761Interactions of substrates and phosphinyl containing inhibitors with bacterial and human and zinc proteasesPLOS ONE

Dear Dr. Sylte,

Thank you for submitting your manuscript to PLOS ONE. After careful consideration, we feel that it has merit but does not fully meet PLOS ONE’s publication criteria as it currently stands. Therefore, we invite you to submit a revised version of the manuscript that addresses the points raised during the review process.

We look forward to receiving your revised manuscript.

Kind regards,

Yash Gupta, Ph.D.

Academic Editor

PLOS ONE

Journal Requirements:

2. Please ensure that you refer to Figure 9, 10, 11 in your text as, if accepted, production will need this reference to link the reader to the figure.

Additional Editor Comments :

None of the reviewer's have recommended any supplementary experiments; therefore, the manuscript revision falls into the minor revisions category. A point-wise address/rebuttal is required.

Reviewers' comments:

Reviewer's Responses to Questions

**Comments to the Author**

1. Is the manuscript technically sound, and do the data support the conclusions?

Reviewer #1: Partly

Reviewer #2: Yes

Reviewer #3: Yes

2. Has the statistical analysis been performed appropriately and rigorously? 

Reviewer #1: N/A

Reviewer #2: Yes

Reviewer #3: Yes

3. Have the authors made all data underlying the findings in their manuscript fully available?

Reviewer #1: Yes

Reviewer #2: Yes

Reviewer #3: Yes

4. Is the manuscript presented in an intelligible fashion and written in standard English?

Reviewer #1: Yes

Reviewer #2: No

Reviewer #3: Yes

5. Review Comments to the Author

Reviewer #1: The manuscript need major revsion before it can be recommended for publication. There are some point which authors needs to improve:

1. The abstract effectively summarizes the experimental and computational results, but it lacks a clear statement of the study’s objectives and significance. For example, explicitly state why comparing bacterial vs. human zinc proteases with these inhibitors is important.

2. The introduction covers relevant background on metalloproteases and zinc-binding groups, However the scientific gap and the motivation for the study still need attention. For example, explain why it is important to study both bacterial (“thermolysin-like”) and human (MMP) zinc proteases together, and why phosphinyl inhibitors are of particular interest.

3. The introduction also mention “novel substrate ES005” without much explanation. It would help to briefly note why ES005 was chosen and what is expected, addressing what particular group in this substrate are of interest. Authors can follow https://pubs.rsc.org/en/content/articlelanding/2022/nj/d2nj02482a for stating why paerticular substrate/compound is needed and what could be expected from this.

4. The methods are detailed, but the number of experimental replicates and statistical treatment should be specified. For example, for enzyme assays (Km and Ki determinations): how many independent experiments were performed and how error was estimated. Also for molecular modelling authors have not mentioned the grid size of the active site. Authors can follow https://onlinelibrary.wiley.com/doi/full/10.1002/jmv.29594 to improve the material and method section for molecular modelling.

5. The term “residual interaction fraction” is introduced in the MD discussion but not explained. Either define this metric (percentage of time a bond is present?). Readers unfamiliar with this phrasing may find it confusing.

6. The Conclusions summarize the findings well, but need to improve. For instance, explicitly mention how these results advance the understanding of inhibitor selectivity or could guide the design of new antibacterials. Right now the conclusion is mostly a recapitulation of results. Highlighting one or two key takeaways for future work would strengthen it.

Minor Comments

"we have not demonstrated that bacterial en zymes prefer ES005 over ES001 and that human MMPs favour ES001 over ES005” is somewhat confusing. If prior studies never directly compared the two substrates, rephrase more clearly

Some sentences are lengthy and could be split or rephrased

The conclusion states that H-3 to H-5 had Ki values in the “high range.” It would be clearer to say explicitly that “H-3, H-4, and H-5 did not show appreciable inhibition (Ki >100 μM) for any enzyme.

Reviewer #2: The studies reported in the manuscript including the enzyme kinetics and the in silico simulations were rigorous and supports the conclusion. The methods were discussed in a comprehensive and reproducible manner. However, the manuscript could be improved by addressing the grammatical and typographical errors. Please address the following comments and general questions and refer to the attached PDF for page specific comments.

1. What is the rationale behind selecting ES001 and ES005 over other available substrates?

2. For the MD simulations, please report the pressure used for the equilibrium.

3. It would help the reader to visualize the data better if S2 Fig is part of the main text of the manuscript.

4. Some supplemental figure numbers referred in the main text are incorrect. Please check the numbers.

5. Please provide the PDB ID of the MMP-9 mutant used in the MD simulations (page 17).

6. The last two sentences of the sub-pocket amino acids paragraph was hard to follow. Please consider adding a comparison of the data from Rahman et al. 2021 to the data reported in Table 3.

7. Please label Fig 7A highlighting the key residues and the sub-pockets. For Fig 7A and 7B, a surface image of the protein complex with labeled sub-pockets would help to visualize and understand the results.

Reviewer #3: This manuscript submitted by Fatema and group explores the molecular interactions of phosphinyl-containing inhibitors with bacterial zinc metalloproteases (TLN, PLN, ALN) and human MMPs (MMP-9, MMP-14), using enzymatic assays, MALDI-TOF MS, and molecular modeling techniques including MD simulations and induced fit docking. The study is methodologically solid and provides insightful data for the development of selective zinc protease inhibitors. The manuscript can be accepted after minor revision mention below

1. The authors used induced fit docking for H-1 and H-2, no comparison with crystallographic data (where available) or experimental validation (e.g., SAR) is provided to confirm pose accuracy

2. Please label measurements for the key interactions (bong length)

3. Only five phosphinyl-containing inhibitors were tested. The conclusions regarding selectivity would be stronger with a broader panel of inhibitors (even with computational predictions).

4. Authors should provide molecular weights and structural notations for clarity.

5. Authors are suggested to rewrite the abstract for more clarity e.g seprate out methods, results and conclusion.

6. PLOS authors have the option to publish the peer review history of their article (what does this mean? ). If published, this will include your full peer review and any attached files.

**Do you want your identity to be public for this peer review?** For information about this choice, including consent withdrawal, please see our Privacy Policy .

Reviewer #1: **Yes: ** SUMIT KUMAR

Reviewer #2: No

Reviewer #3: No

---

## [Author Response · Author response to Decision Letter 1]

7 Jul 2025

Manuscript ID: PONE-D-25-23761

The response and changes according to requirements and comments from academic editor and reviewers are shown in the file "Response to Reviewers"

Comments to the last email from PLos One with the heading "Edits requested on your submission PONE-D-25-23761R1":

1. Figure 9-11 (Figure 10-12 in the present version) are refered to on page 28 in the present version of the manuscript. Please see point 2 under “Journal requirements” in the Response to reviewers letter.

2. A file entitled “Response to reviewers” has been uploaded.

3. In addition raw data files behind Fig. 7 and S8 Fig and the calculations of Ki and IC50 values have been uploaded. Two files in ppt format and 5 files in Excel.

---

## [Editor Report · Decision Letter 1]

Interactions of substrates and phosphinyl containing inhibitors with bacterial and human zinc proteases

PONE-D-25-23761R1

Dear Dr. Sylte,

We’re pleased to inform you that your manuscript has been judged scientifically suitable for publication and will be formally accepted for publication once it meets all outstanding technical requirements.

Kind regards,

Yash Gupta, Ph.D.

Academic Editor

PLOS ONE
---

## [Editor Report · Acceptance letter]

PONE-D-25-23761R1

PLOS ONE

Dear Dr. Sylte,

I'm pleased to inform you that your manuscript has been deemed suitable for publication in PLOS ONE. Congratulations! Your manuscript is now being handed over to our production team.

Kind regards,

on behalf of

Dr. Yash Gupta

Academic Editor

PLOS ONE